# Pennate diatoms make non-photochemical quenching as simple as possible but not simpler

Dany Croteau, Marianne Jaubert ⓘ , Angela Falciatore ⓘ & Benjamin Bailleul ✉

Studies of marine microalgal photosynthesis are heavily moulded on legacy research from organisms like *Arabidopsis* and *Chlamydomonas*, despite the differences between primary and secondary endosymbionts. Non-photochemical quenching (NPQ) protects photosystem II from excessive light and, in pennate diatoms, requires the xanthophyll pigment diatoxanthin and Lhcx proteins. Although NPQ's relationship with diatoxanthin is straightforward, the role of Lhcx proteins has been unclear and at the core of several conflicting NPQ models, often unnecessarily borrowing the complexity of models from green organisms. We use 14 *Phaeodactylum tricornutum* strains, including 13 transgenic lines with variable *Lhcx1* expression levels, grow them under two non-stressful light conditions, and modulate diatoxanthin levels through short light stress. The resulting Lhcx1-diatoxanthin matrices are used to demonstrate that NPQ is proportional to the product of the Lhcx1 concentration and the proportion of diatoxanthin in the xanthophyll pool. This indicates that the interaction between diatoxanthin and Lhcx1 creates a homogeneous Stern-Volmer quencher responsible for NPQ. Additionally, we demonstrate that the photosynthetic unit in pennate diatoms follows a "lake" model, with discrepancies in the NPQ-photochemistry relationship arising from unconsidered assumptions, one possibility being cellular heterogeneity. This underscores pennate diatoms as natural reductionist system for studying marine photosynthesis.

By comparison with plants and green algae, the model pennate diatom *Phaeodactylum tricornutum* is, in many ways, a simplified system to appreciate the constraints and regulatory networks that outline photosynthesis. For instance, its four plastid membranes are loosely arranged and show no heterogeneous lateral segregation of photosynthetic complexes, both photosystem (PS) have similar absorption spectra and typically accumulate at roughly equivalent stoichiometry (see review[1]). Such features combined with growing molecular tools adapted to diatoms, make *P. tricornutum* an ideal model to probe the specific challenges that apply to the oceanic fraction of photosynthesis[2] (≈50% of global primary production[3]), its adaptations

to marine environments (specific light environment, $HCO_3^-$ as the main C source, no hydric stress, etc.) and its plasticity in the context of climate change. Indeed, phototrophs derived by secondary endosymbiosis events (mainly diatoms, haptophytes, and dinoflagellates), and not primary endosymbionts of the green lineage, dominate photosynthesis in the Ocean. Strikingly, the main photosynthetic complexes are exceptionally well conserved across organisms and the universal goal of photosynthesis is to match light harvesting with NADPH/ATP demands in dynamic environments. Yet, to adapt to niche-specific challenges and fuel original metabolic pathways[4,5], a plethora of regulatory pathways and antenna proteins evolved across

CNRS, Sorbonne Université, Institut de Biologie Physico-Chimique, Photobiologie et physiologie des plastes et des microalgues - P3M, Paris, France. ✉e-mail: bailleul@ibpc.fr

groups[6]. Still how this diversity affects functions remains often overlooked as, rich in more than a century of research on *Arabidopsis*, common crops, *Chlamydomonas* and *Chlorella*, the "green paradigm" chiefly rules how photosynthesis experiments are conducted and interpreted regardless of the (eukaryotic) organism studied.

This is nicely exemplified by the case of non-photochemical quenching (NPQ). By increasing the proportion of PSII chlorophyll (Chl *a* de-excitation through heat-releasing pathways, NPQ is crucial to limit risks of generating destructive oxygen species leading to PSII photoinhibition and thus, deeply influence global carbon fluxes[7]. While NPQ is observed in virtually all phototrophs, it arises from collections of mechanisms, governed by a multitude of different molecular players and varying with acclimation conditions[6]. Several mechanisms are known to affect PSII fluorescence and muddle NPQ interpretation in plants, red algae, and cyanobacteria, like state-transition[8], chloroplast movements[9], and PSII-to-PSI spillover[10]. Those mechanisms appear absent in *P. tricornutum* and all diatoms so far[11]. Moreover, in contrast to the compounded nature of NPQ in most organisms, *P. tricornutum*'s NPQ relaxes as a mono-exponential decay in darkness with a lifetime of ≈10 min (if the quenching associated with PSII damages (qI) is avoided). In pennate (and some centric) diatoms, NPQ is linearly correlated to the de-epoxidized xanthophyll pigment diatoxanthin (DT)[12–18]. Diatoms possess a single-step xanthophyll cycle (XC) which converts diadinoxanthin (DD) into DT[19], versus the two-step XC in other organisms[20,21]. The proportion of the xanthophyll pigments active in NPQ (i.e., DT) is called de-epoxidation state (DES) and is calculated as (DT/(DD + DT)).

These features suggest that NPQ in *P. tricornutum* is a very simple process. The linear relationship between NPQ and DT resembles the behaviour of a homogeneous Stern-Volmer (SV) quencher *Q* in solution, where the extent of fluorescence quenching is expressed as its concentration [*Q*] multiplied by its quenching rate constant ($\kappa_{SV}$). In photosynthesis, the "lake" model defines a photosynthetic unit containing numerous PSII complexes embedded in a common light-harvesting antenna and competing for excitons[22,23]. The yield of each process involved in excited Chl *a* relaxation (fluorescence, heat dissipation, photochemistry or NPQ) is then determined by the kinetic competition between their rate constants, the one of photochemistry being proportional to the concentration of open reaction centres with oxidized $Q_A$ ([$Q_A$])[24,25] (see Supplementary Text 1 for details). Therefore, an SV-quencher in a lake model (hereafter called "SV-lake" model), predicts two equations relating NPQ to [*Q*] (Eq. 1) and to the photochemical yield of PSII (Eq. 2):

$$NPQ = F_M/F_{M'} - 1 = \kappa_{SV}[Q] \qquad (1)$$

where $F_M$ and $F_{M'}$ are maximal fluorescence in darkness, before and following light perturbation, respectively.

$$F_{V'}/F_{M'} = F_V/F_M \times (1 - f(NPQ)),$$
$$\text{with} f(NPQ) = \frac{(1 - F_V/F_M) \times NPQ}{(1 + (1 - F_V/F_M) \times NPQ)} \qquad (2)$$

Where $F_{V'}/F_{M'}$ and $F_V/F_M$ are the maximum yield of PSII in the dark ([$Q_A$] = 1), with and without NPQ, respectively, and where *f* (NPQ) represents the expected relative decrease in $F_{V'}/F_{M'}$ for a given value of NPQ. Considering that functional absorption cross-section (σPSII) and maximum quantum yield of PSII vary together and remain proportional as heat dissipation processes take place in the antenna[26], the relationship between σPSII and NPQ should also follow Eq. 2 (see Supplementary Text 1). The lake model is a theoretical simplification, in practice, photosynthetic samples are composed of non-connected domains (e.g., different cells, different thylakoids within a cell). However, multiple domains do not necessarily invalidate the lake model provided all domains in the sample behave as a "lake" and share

homogeneous properties/parameters ([$Q_A$], [$Q$] and de-excitation pathways' rate constants). Many attempts were made for more precise, and complex, models, intermediate between the "lake" model and the "puddle" model which considers that each PSII has its own independent light-harvesting system[22,23]. Yet, the lake model provides immense descriptive and predictive power despite its simplicity and is used to derive a multitude of photosynthetic parameters (see Supplementary Text 1).

Several groups tested the "SV-lake" model in *P. tricornutum*. Although the linear relationship between NPQ and DT (Eq. 1) still holds, deviations from theory were reported regarding the relationship between NPQ and PSII photochemistry (Eq. 2). In ref. 12, the authors measured a larger than predicted non-photochemical quenching of minimal fluorescence ($F_0'$, open PSII reaction centres), relative to the non-photochemical quenching of maximal fluorescence ($F_{M'}$, closed reaction centres) (see Supplementary Text 1). In ref. 27, a higher than predicted $F_{V'}/F_{M'}$ for a given NPQ, or an apparent "excess of PSII photochemistry", was reported. In both scenarios, the efficiency of the non-photochemical quencher appeared higher when reaction centres were open rather than closed, a pattern previously coined "economic quenching" in plants[28]. More recently, higher than predicted σPSII for a given NPQ was similarly interpreted as an "antenna uncoupling" model[29], another model inspired by plants[30,31], but intrinsically incompatible with an SV-lake model. Depending on models, the functional uncoupling of the PSII antenna would either entirely explain Chl *a* fluorescence decrease[29], or act as the main mechanism among two quenching processes[27,32,33]. Due to the expected proportionality between σPSII and $F_{V'}/F_{M'}$, the different observations (both σPSII and $F_{V'}/F_{M'}$ higher than predicted) likely indicate a shared phenomenon (see Supplementary Text 1) that seems a priori inconsistent with the SV-lake model.

Another challenge concerns the nature of the SV-quencher, which cannot be DT *alone* since the slope of the NPQ vs. DT relationship is highly variable between growth conditions[13,15,16,34–36]. Two explanations have been proposed, which are not mutually exclusive. First, a low luminal pH could "activate" DT by protonating the acidic residues of some protein partners[14,37], similar to the role of PsbS in plants[38] or LHCSR3 in *Chlamydomonas*[39]. Second, variations in the slope may arise from differences in the proportion of xanthophylls associated with the PSII antenna. Indeed, other xanthophyll pools have been described, including one soluble in the lipid phase and one associated with PSI[35,36,40]. Another argument against the original idea that DT *alone* acts as an SV-quencher[12,41] is the strict requirement of a second molecular effector for NPQ: Lhcx proteins[42,43], which are LHC stress-related subfamily close to LHCSR3[44]. Each diatom species relies on a distinct array of Lhcx isoforms. *Phaeodactylum tricornutum* possesses four of them. Lhcx1-2-3 influences NPQ levels[29], but only Lhcx1 is constitutively expressed[42]. Unlike in plants, where high-energy quenching (qE) can occur in the absence of PsbS or in the absence of zeaxanthin[21], both Lhcx proteins and DT are mandatory for the reversible NPQ in *P. tricornutum*[29].

With PSII fluorescence as the sole observable, all models and hypotheses presented above are doomed to self-referential dead-ends when it comes to determining whether apparent incongruities convey information about "real" photophysiological features or about the validity of the "SV-lake" model. Pennate diatoms could offer the exceptional opportunity to test the coherence of the "SV-lake" model via a second observable, [*Q*], provided its nature can be elucidated and measured. Due to its simplicity and predictive power grounded in established physical principles, identifying a natural system for which the "SV-lake" model applies without incoherencies would be a pivotal step in photosynthesis research. To strive towards these goals, identifying the nature of *Q* and testing the "SV-lake" model, we introduce an approach involving "molecular titration" of Lhcx1. We generated 13 transgenic strains which, together with *P. tricornutum* wildtype, can be

seen as a continuum of Lhcx1 concentration rather than discrete entities. These strains were grown under two non-stressful growth conditions, leading to the expression of the Lhcx1 isoform only. We modulated the amount of DT with brief light stress, allowing us to explore NPQ (and other parameters such as σPSII or $F_V'/F_M'$) in two Lhcx1×DES matrices defined by each growth condition. These data allowed us to establish robust phenomenological relationships between NPQ, DT, and Lhcx1, identify a promising candidate for the homogeneous SV-quencher, and revisit the validity of the "SV-lake" model in pennate diatoms.

## Results

### Varying Lhcx1 and diatoxanthin concentrations, everything else equal

Both Lhcx1 and DT influence NPQ levels[12,29], but their respective concentrations have never been modulated and quantified simultaneously under controlled conditions to develop a simple and comprehensive model unifying their respective involvement in NPQ. To do so, we used a "molecular titration" approach by cultivating wildtype (WT) *P. tricornutum* (Pt2, CCAP 1052/1 A) and 13 transgenic strains (Lhcx1-KO and 12 Lhcx1-complemented (described in ref. 45)), under two non-stressful growth conditions known to be associated with contrasted XC pigments and Lhcx1 accumulations (Fig. 1). Other isoforms are not or barely expressed in the absence of high light periods[42,45] or nutrient deprivation[46]. We first measured Lhcx1 accumulation in the 14 strains grown under intermittent light (IL) conditions using Western blots (Fig. 1a), as we had previously done for these strains grown under the low light (LL) conditions (first reported in ref. 45). Normalising Lhcx1 accumulation to its maximal value (IL-grown complemented strain LtpM, for *Lhcx1* Talen target site modified plasmid, strain M) provided a relative [Lhcx1] scale encompassing all strains and growth conditions, used for the colour code in the figures (see "Methods" section). [Lhcx1] ranged from 0 to 0.37 ± 0.02 under LL and from 0 to 1 under IL, confirming larger pools of Lhcx1 under IL than LL. The WT displayed intermediate values of 0.07 ± 0.03 under LL, and 0.68 ± 0.11 under IL conditions (Fig. 1b/c and Supplementary Fig. 1).

Then, we tested whether inter-strains baseline physiology was influenced by [Lhcx1]. Overall, under LL and IL, linear regression analysis indicated that broad changes in [Lhcx1] did not significantly influence ($p$.value > 0.05) baseline physiological parameters (Table 1) like growth rates (only measured in LL), the maximum quantum yield ($F_V/F_M$) and functional absorption cross-section (σPSII) of PSII, and the light-harvesting pigments composition (Fig. 1d–g). However, $F_V/F_M$, σPSII and the content of fucoxanthin and Chl $c$, relative to Chl $a$, were all significantly higher (10–20%) under LL than IL ($p$.value < 0.01, One-Way ANOVA) (Fig. 1 and Supplementary Fig. 2). Unexpectedly, photoprotective pigments (xanthophylls and β-carotene) correlated significantly with [Lhcx1], which resulted in a significant increase of the xanthophyll-to-β-carotene ratio, by ≈50% under LL and ≈100% under IL, from lowest to highest [Lhcx1] ($p$.value < 0.01, Table 1 and Fig. 2b). This expands the space within which we can test NPQ models, as we have two independent Lhcx1-DES matrices associated with distinct PSII characteristics and light-harvesting antenna. Nevertheless, for each growth condition, baseline physiology was largely unaffected by [Lhcx1], allowing us to explore variations of Lhcx1 and/or DT (using different light protocols, see below), everything else being roughly equal (all data found in Supplementary Data 1).

### Relationships between NPQ, de-epoxidation state and Lhcx1

We first measured NPQ, using a fluorescence imaging system, and xanthophyll pigments by HPLC under steady-state conditions at different light intensities and at different relaxation times after a shift from high light to darkness (see "Methods" section and see Supplementary Fig. 3 for examples of NPQ measurements). To compare

treatments with varying xanthophyll pools (Fig. 2), we correlated NPQ to DES, but all relationships with DT are found in Supplementary Figs. 4, 6 and 7. When NPQ relaxation was followed, we observed that the fully relaxed $F_M'$ returns to a value very close to the initial dark-adapted $F_M$ (Supplementary Fig. 5a, b). This confirmed that only a fast-relaxing component of NPQ was significant in our experiments, allowing us to neglect contributions from photoinhibition-related quenching (qI) in our interpretation of the NPQ data.

As a function of light intensity, NPQ and DES followed sigmoidal curves[47] (Fig. 3a–d). As expected, larger [Lhcx1] resulted in larger capability to deploy NPQ among strains, and the relationship between maximal NPQ (NPQ_M, see "Methods" section) and [Lhcx1] (Table 1) scaled near-perfectly across both growth conditions ($R^2 = 0.97$) (Fig. 2a). The other fitted parameters, the light intensity for half-saturation (E50NPQ) and the sigmoidal coefficient ($n$), also varied significantly ($p$.value < 0.05) with [Lhcx1] (Table 1 and see "Methods" section). All strains appeared to converge towards a similar DES plateau of 40% for LL (Fig. 3c). Under IL, most strains did not reach saturated DES value at the highest light intensity, with only Lhcx1-KO stabilising around 60% (Fig. 3d). Unsurprisingly, NPQ was strongly linearly correlated with DES (Fig. 3e, f) and DT (Supplementary Fig. 4) for all strains and growth conditions; however, with very different slopes depending on their [Lhcx1]. Specifically, the slope of the NPQ vs. DES relationship was proportional to [Lhcx1] regardless of growth conditions (Fig. 3g). The robust relationship between NPQ, [Lhcx1] and DES, across strains, light intensities and growth conditions (despite significant variations between LL and IL PSII characteristics, see Fig. 1 and Supplementary Fig. 2) prompted us to test its stability during transitory phases. Indeed, this relationship was maintained during NPQ relaxation in darkness for LL strains (Supplementary Figs. 5, 6 and open symbols in Fig. 3g, h). For IL strains, NPQ relaxation in darkness was faster than in LL, and the NPQ/DES relationship deviated from strict proportionality for some strains, showing a trajectory that did not pass through the origin of the axes. This complicated the linear fit (see Supplementary Fig. 6 and "Discussion" section). Overall, Lhcx1 appears to act as an NPQ enhancer for a given DES, so that NPQ could be expressed as 0.25 × [Lhcx1] × DES in our relative scale (Fig. 3h). Conversely, no strong relationship encompassing both growth conditions was found between NPQ, [Lhcx1] and DT (Supplementary Fig. 7).

The phenomenological relationship between NPQ, [Lhcx1] and DES, remains robust across 9 strains, six light intensities, steady-state and transitory phases, and two different growth conditions (Fig. 3h). It supports the concept of a homogeneous Stern-Volmer non-photochemical quencher ($Q$), whose quenching coefficient is consistent across the two light conditions (see "Discussion" section) and whose concentration corresponds to [Lhcx1] × DES for the Lhcx1 quantification used here. This conserved relationship provides significant insight into the nature and molecular mechanism of NPQ. First, it strongly argues against a composite nature for NPQ in *P. tricornutum* in contrast to *Chlamydomonas* or *Arabidopsis*[6]. It is hardly reconcilable (see "Discussion" section) with complex NPQ models involving two quenching sites (e.g., refs. 27,32,33), direct pH regulation of LHCSR3/PsbS as in green organisms[38,39] or PSII antenna uncoupling[29]. Instead, this robust relationship supports the previous proposal[48] that a single component dominates rapidly reversible NPQ in pennate diatoms. This NPQ component being strictly proportional to the proportion of de-epoxidized xanthophyll, it is analogous to the strictly zeaxanthin-dependent and slowly relaxing component qZ, observed in *Arabidopsis* PsbS mutants[21,49] (see "Discussion" section). Overall, these observations are compatible with a model where the SV-quencher $Q$ is generated by DT interacting with Lhcx1. Assuming each Lhcx1 protein interacts with DD/DT (possibly via binding see ref. 50 and "Discussion" section) and that only the Lhcx1-DT complex generates $Q$ and induces NPQ/qZ, then [$Q$] would be the product of [Lhcx1] (the concentration

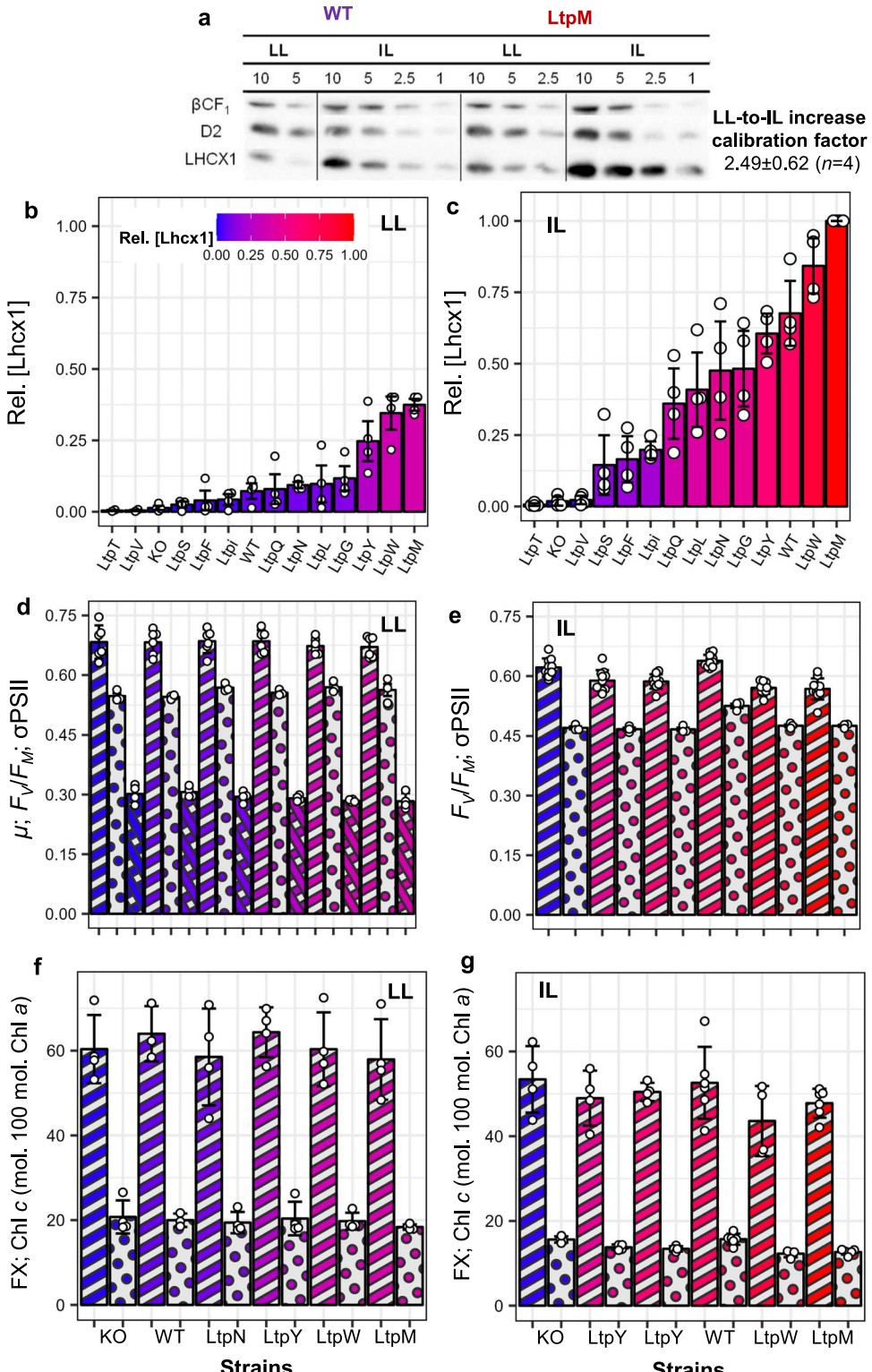

**Fig. 1 | Lhcx1 accumulation and baseline physiology of 14 *Phaeodactylum tricornutum* strains cultivated under two different light conditions.** Western blots on wildtype (WT) and Lhcx1-complemented strain LtpM, with different protein loadings (1–10 μg), were used to calculate changes in [Lhcx1] normalised to the sum of ATP synthase βCF1 and PSII-D2 subunit (see "Methods" section), between low light (LL) and intermittent light (IL) conditions (**a**). The obtained scaling factor allowed us to calculate a relative [Lhcx1] scale common to LL (**b**) and IL (**c**) growth conditions, shown on a 0-to-1 colour-scale (see text). Full blots for IL are shown in Supplementary Fig. 1 whereas relative Lhcx1 quantifications in LL are taken from ref. 45. Baseline physiology parameters were measured under LL (**d**) and IL (**e**) on six strains exemplifying the [Lhcx1] gradient, including the dark-acclimated quantum yield of PSII ($F_V/F_M$, single stripped bars), growth rate (μ, d$^{-1}$, X-stripped bars) and PSII functional absorption cross-section (σPSII, in Å$^2$·10$^{-3}$, dotted bars). Light-harvesting pigment composition was also measured under LL (**f**) and IL (**g**) (full dataset in Supplementary Data 1 and Supplementary Fig. 2), including fucoxanthin (FX) (single stripped bars) and chlorophyll (Chl) *c* (dotted bars) normalised to Chl *a*. Bars height represent mean ± SD and white dots represent all independent biological replicates measured. Linear regression results for mean parameters value vs. [Lhcx1] are in Table 1.

**Table 1 | Lhcx1 concentration dependence of various physiological parameters in *Phaeodactylum tricornutum***

| Physiological process | Parameter | Growth condition | Slope | y-intercept | $R^2$ | p.value | N |
|---|---|---|---|---|---|---|---|
| Growth | μ | LL | −0.03 | 0.30 | 0.22 | 0.09 | 14 |
| PSII Fluorescence | $F_V/F_M$ | LL[c] | −0.01 | 0.68 | 0.05 | 0.45 | 14 |
| | | IL | −0.02 | 0.61 | 0.16 | 0.15 | 14 |
| | σPSII | LL[c] | −0.44 | 563.61 | 1.58E-05 | 0.99 | 14 |
| | | IL | 21.51 | 465.43 | 0.11 | 0.46 | 7 |
| Light-harvesting pigments | FX/Chl a | LL[c] | −4.12 | 61.70 | 0.05 | 0.66 | 6 |
| | | IL | −3.49 | 50.72 | 0.15 | 0.31 | 9 |
| | Chl c/Chl a | LL[c] | −3.37 | 20.42 | 0.38 | 0.19 | 6 |
| | | IL | −1.38 | 14.38 | 0.17 | 0.28 | 9 |
| Photoprotective pigments | XP/Chl a | LL | 9.94 | 6.34 | 0.85 | 0.01[b] | 6 |
| | | IL[c] | 5.37 | 15.31 | 0.91 | 5.59E-05[b] | 9 |
| | β-car./Chl a | LL | −1.08 | 4.39 | 0.61 | 0.07 | 6 |
| | | IL | −1.34 | 5.00 | 0.68 | 0.01[b] | 9 |
| | XP/β-car. | LL | 2.94 | 1.43 | 0.81 | 8.75E-04[b] | 6 |
| | | IL[c] | 2.74 | 2.92 | 0.91 | 3.28E-03[b] | 9 |
| NPQ induction | $NPQ_M$ | LL | 13.20 | 0.38 | 0.97 | 1.69E-10[b] | 14 |
| | | IL | 11.97 | 0.38 | 0.96 | 4.53E-10[b] | 14 |
| | E50NPQ | LL | 275.64 | 133.86 | 0.44 | 0.02[b] | 11[a] |
| | | IL | 433.48 | 193.85 | 0.86 | 4.07E-05[b] | 11[a] |
| | nNPQ | IL | −2.07 | 3.87 | 0.82 | 1.26E-04[b] | 11[a] |
| NPQ vs. XP (induction) | a(NPQ/DES) | LL | 0.27 | 0.01 | 0.97 | 4.34E-04[b] | 6 |
| | | IL | 0.21 | 0.12 | 0.91 | 2.93E-03[b] | 9 |
| | a(NPQ/DT) | LL | 2.67 | 0.22 | 0.98 | 4.48E-07[b] | 6 |
| | | IL | 1.05 | 0.10 | 0.95 | 7.46E-06[b] | 9 |
| NPQ vs. XP (relaxation) | a(NPQ/DES) | LL | 0.27 | 0.18 | 0.98 | 1.27E-04[b] | 6 |
| | | IL | 0.16 | 0.19 | 0.90 | 8.01E-05[b] | 9 |
| | a(NPQ/DT) | LL | 2.45 | 0.23 | 0.91 | 3.24E-03[b] | 6 |
| | | IL | 0.74 | 0.15 | 0.87 | 2.59E-04[b] | 9 |

Outcomes of linear regression analysis (two-sided) for various physiological parameters vs. [Lhcx1], in *Phaeodactylum tricornutum* wildtype and Lhcx1-mutant strains cultivated under either low light (LL) or intermittent light (IL) growth conditions, where *N* is the number of strains accessed for a given parameter. A two-sided ANOVA test was also carried out between growth conditions and reported in Supplementary Fig. 2. The growth rate (*μ*) for IL strains was not measured, xanthophyll pigments (XP) comprises diadinoxanthin (DD) plus diatoxanthin (DT), the de-epoxidation sate (DES) is calculated as DT/(DD + DT), the sigmoidicity parameter for non-photochemical quenching (NPQ) induction (*n*) is globally fitted across all strains for LL (see Supplementary Data 1) so its dependency to [Lhcx1] could not be tested, details on total biological replicates and mean ± SD for each strain and growth conditions are found in Supplementary Data 1.

$F_V/F_M$ dark-acclimated maximal quantum yield of PSII, *σPSII* PSII functional cross-section, *FX* fucoxanthin, *β-car.* β-carotene, $NPQ_M$ fitted extrapolated maximum NPQ, *E50NPQ* light intensity for half-saturation of NPQ.

[a]For three strains with relative [Lhcx1]=0, NPQ fitted parameters were removed from the linear regressions because NPQ induction was null.

[b]Indicate significant *p*.value (<0.05) for the linear regression analysis.

[c]Indicate a significantly lower value (*p*.value<0.05) of a given parameter under this growth condition as per the ANOVA test.

of xanthophyll interaction sites) and DES (the proportion of xanthophylls in the DT form) (Fig. 3i).

**Relationships between photochemistry and NPQ and the SV-lake model**

Given compelling results that NPQ/qZ behaves like an SV-quencher proportional to [Lhcx1] × DES in *P. tricornutum*, we decided to revisit the "SV-lake" model. This model predicts a unique relationship (Eq. 2) between the potential quantum yield of PSII for photochemistry in the dark ($F_V/F_M$) and NPQ level. Yet, previous studies have shown a consistent deviation from this relationship in *P. tricornutum*, indicating excess photochemistry (or excess σPSII) for a given NPQ[12,27,29]. We induced NPQ/qZ under 6 min of high light followed by concomitant monitoring of NPQ/qZ relaxation and $F_V'/F_M'$ recovery in darkness. The final part of this work focuses specifically on the relaxation of the NPQ in the dark, which allows us to mathematically separate the fast-relaxing component of NPQ from the slow-relaxing one. We achieved this by using, as a reference, the $F_M'$ value at the end of the fast relaxation phase (see "Methods" section), which was not possible in the first part of this work relying partly on steady-state NPQ values. To

clarify that any minor contributions from slow-relaxing components of the NPQ, such as qI, are now excluded, we will henceforth refer to the fast-relaxing component as qZ rather than the more general and less precise term NPQ.

For comparison with previous studies, we focused on cultures acclimated to moderate light (ML) (40 μmol photons $m^{-2} s^{-1}$, 12:12 L:D) but LL and IL conditions were also investigated (Supplementary Figs. 8 and 9). For ML-acclimated experiments, we selected the WT and six strains showing higher NPQ capacity than the WT under LL. The qZ reached values ranging from 2 to 4 between strains at light offset, and subsequently relaxed in darkness with a half-time of approximately 7 min (Fig. 4a and Supplementary Fig. 7). To test the SV-lake model, we plotted the measured $F_V'/F_M'$ as a function of the expected relative decrease in $F_V'/F_M'$ due to qZ, given by the parameter *f* (qZ) (instead of *f* (NPQ) in Eq. 2, see "Methods" section). Like others before us[12,27], we observed that $F_V'/F_M'$ in the early phase of qZ relaxation was larger than predicted by the SV-lake model (Fig. 4b). Thus, plotting $F_V'/F_M'$ against *f* (qZ), which represents the expected relative decrease of $F_V'/F_M'$ for a given qZ in the SV-lake model (see Eq. 3 in "Methods" section), produced a linear

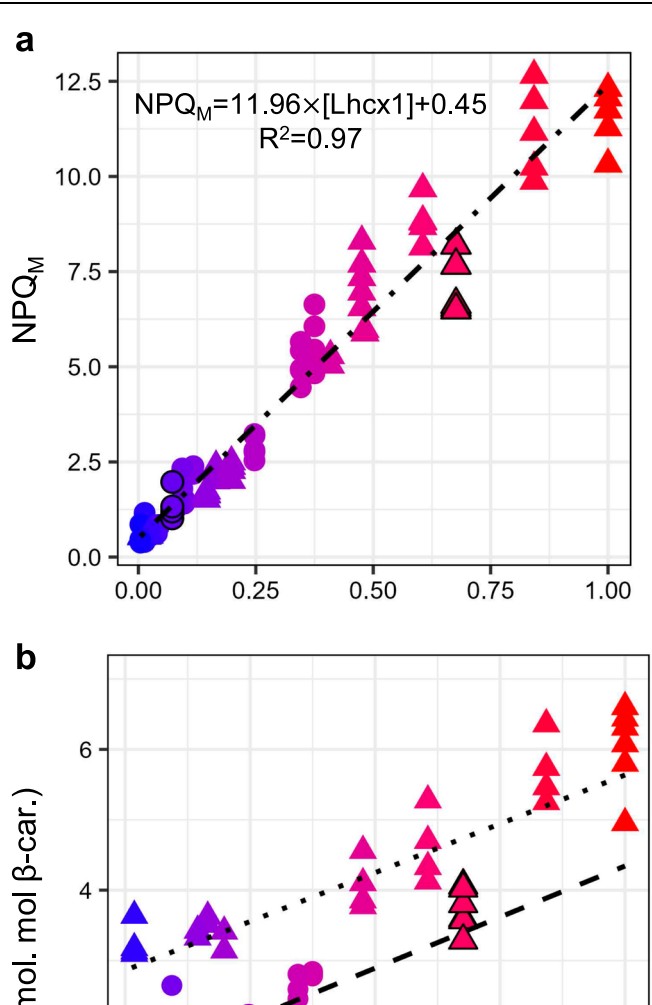

**Fig. 2 | Lhcx1 dependence of maximal NPQ capacity and xanthophyll pigments pool.** Linear regression of extrapolated maximum NPQ ($NPQ_M$) (see "Methods" section) vs. [Lhcx1] across both growth conditions (LL, circles; IL, triangles), with the same colour code that is used in Fig. 1. The wildtype (WT) *Phaeodactylum tricornutum* (black outline symbols) and all 13 Lhcx1 mutants are shown (**a**). All biological replicates of xanthophyll pigment (XP) pool normalised to β-carotene (see "Methods" section) vs. [Lhcx1] for both growth conditions (**b**). All replicates measured on independent biological replicates are shown, and the linear regressions are fitted on mean values per strain × growth conditions which can be found in Supplementary Data 1.

relationship ($R^2 = 0.97$) but, a slope of −0.68 rather than predicted −1 (Fig. 4c). We calculated the Ratio of experimental-to-theoretical $F_V'/F_M'$, $R_{exp/th}$ (see "Methods" section); it generally increased from 1 to 3 min after the dark transition, before reversing concomitantly with qZ relaxation (Fig. 4b). The same trend was observed in strains acclimated to LL and IL (Supplementary Figs. 8 and 9), but with smaller gap between experimental and theoretical values than in ML. Negative $R_{exp/th}$ values were also sometimes observed in the first phase of the dark transition, which we interpret as PSII

photochemical yield being limited by incomplete reoxidation of $Q_A^-$ (Supplementary Figs. 8 and 9).

Having replicated observations made by others, we considered three possible explanations for this deviation from the theory. In addition to the "PSII antenna uncoupling" hypothesis[29] and the "economic quenching" hypothesis[27], we introduced a third one, the "heterogeneity" hypothesis, which suggests that the SV-lake model is valid locally, in each domain/cell, but with different properties between domains. We hypothesised that the apparent "excess" PSII photochemistry might result from averaging many "lake" domains with varying properties, such as [Lhcx1] or $F_V/F_M$ values (see full empirical demonstration in Supplementary Text 2). Such heterogeneity can either exist between PSII sub-populations within a cell[51] or between cells. Heterogeneity between cells is custom in microalga cultures and one way it can be visualised is by using flow cytometry and DAPI staining to measure DNA/cell and cell biovolume as markers of cell cycle phase distribution[52] (Fig. 4d).

### NPQ/qZ and PSII antenna uncoupling

We first examined the "antenna uncoupling" hypothesis. We used a similar protocol but with a fluorometer based on single turnover flashes as in ref. 29, allowing the monitoring of both PSII functional cross-section (σPSII') and $F_V'/F_M'$ in parallel to qZ relaxation, (see "Methods" section). Consistent with[29], we observed that σPSII' decreased less than expected for a given qZ level (Fig. 5a). As noted previously, variations in σPSII' can indicate changes in the optical cross-section, i.e., the physical size of PSII antenna (i.e., "antenna uncoupling"[29]), changes in $F_V'/F_M'$, or both. Additionally, we tracked the evolution of $F_V'/F_M'$ relative to qZ and found a similar excess for photochemistry (Fig. 5b). Throughout qZ relaxation, σPSII' remained proportional to $F_V'/F_M'$ (Fig. 5c), implying a constant *optical* cross-section of PSII. This rejects the "PSII antenna uncoupling" hypothesis where NPQ stems (at least part of it) from changes in PSII optical cross-section due to physical uncoupling[27,29,32,33]. Furthermore, it indicates that deviations measured for both observables, σPSII' and $F_V'/F_M'$, are synonymous ways of reporting the same phenomenon, whatever its underlying cause might be.

### The effect of heterogeneity on the relationship between photochemical yield and NPQ/qZ

At this point, we chose to explore the "heterogeneity" hypothesis. To simulate heterogeneity, we selected values compatible with our cytometry (Fig. 4d) and fluorescence measurements, as well as a level of heterogeneity in maximal qZ and $F_V/F_M$ consistent with the single cell fluorescence reported in ref. 53 (Supplementary Text 2). The very good agreement between the fitted relationship and the experimental data (green line in Fig. 4c) demonstrates that intercellular heterogeneity can elucidate the deviation from the SV-lake model, without invaliding it.

To effectively distinguish between "economic quenching" and the "heterogeneity" hypotheses, we reasoned that "economic quenching" should be an inherent property of Q, independent of measurement protocols. In contrast, under the "heterogeneity" hypothesis, one would expect the deviation to vary depending on the degree of heterogeneity, which can be manipulated to some extent. A well-known source of heterogeneity is asynchronicity among cells in a given culture. A conventional protocol to synchronise cells involves incubating cultures in darkness for an extended period (40 h in *P. tricornutum*[52]), thereby reducing inter-cell heterogeneity as all cells complete their final cycle before reaching the same G1 phase. Indeed, flow cytometry and DAPI staining assays revealed clearly more uniform distribution in biovolume and cellular DNA after 40 h of dark incubation than for cultures sampled directly under ML (Figs. 4d and 6d). The same qZ relaxation experiment as under ML was performed after 40 h of dark acclimation, 8 times with the same 7 strains. The reproducibility of the experiments after dark acclimation was lower than in ML, but in all 8 repetitions performed, a linear relationship was found ($R^2 = 0.99$) with

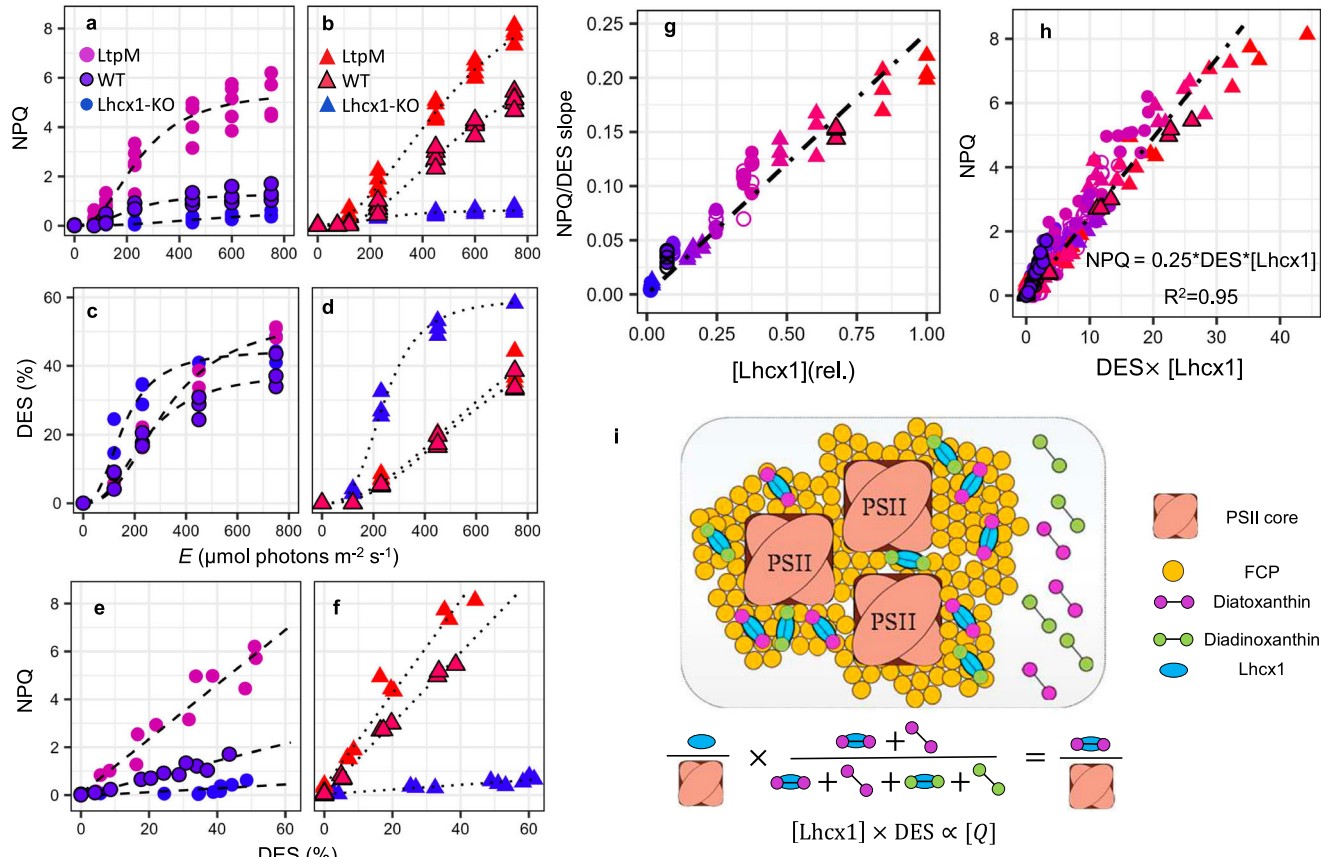

**Fig. 3 | Relationships between NPQ, Lhcx1 and the de-epoxidation state (DES) support an "SV-lake" model in *Phaeodactylum tricornutum*.** The same colour code as in Figs. 1–2 is used. NPQ vs. light intensity (*E*) (**a**, **b**) and DES vs. *E* relationship (**c**, **d**) in *P. tricornutum* wildtype (WT, black outline symbols), KO and the Lhcx1 overexpressor strain, LtpM grown under low light (LL, circle symbols) (**a**, **c**) and intermittent light (IL, triangle symbols) (**b**, **d**). Based on the first 4 panels, the NPQ vs. DES relationship was plotted for LL (**e**) and IL (**f**). NPQ vs. DES slope determined in 9 strains at different steady-state light intensities (closed circle (LL) and triangle (IL) symbols with the same colour code as used in Fig. 1), same protocol as for **e**, **f**) and during relaxation in the dark for 6 LL-grown strains (open symbols,

see "Methods" section) are shown as a function of [Lhcx1] (**g**). For all 291 data points corresponding to experiments shown in (**a**–**g**) (same symbol code), are used for the linear regression of NPQ vs. DES × [Lhcx1] plotted in (**h**). Schematic representation of a working model wherein DT interacting with Lhcx1 is proportional to the concentration of a Stern-Volmer quencher (*Q*) for pennate diatoms' NPQ (**i**). The NPQ vs. DES (and DT) slopes for all strain × growth conditions in steady-state illumination are found in Supplementary Fig. 4; the ones for the dark relaxation protocols in Supplementary Figs. 5 and 6, and the equivalent of (**g** and **h**) when using DT instead of DES to access quenching efficiency in Supplementary Fig. 7. Details on mean ± SD and number of biological replicates are found in Supplementary Data 1.

slopes closer to the theoretical one than before dark acclimation (Supplementary Fig. 9). Even more compelling was the observation that in four of the eight experiments (shown in Fig. 6c), the data showed an almost perfect agreement with the model, encompassing all seven measured strains. This demonstrates that more homogeneous cultures (see all cytograms in Supplementary Fig. 10) tend to deviate less from the SV-lake model and, that varying quenching efficiency between open or closed PSII centres is not an intrinsic feature of qZ.

Finally, we examined the SV-lake model using another pennate diatom, the araphid pennate *Plagiostriata* sp. (formerly *Leptocylindrus danicus*[54]). When acclimated to ML and subjected to the same protocol previously used on *P. tricornutum*, *Plagiostriata* sp. displayed qZ of approximately 2. During dark relaxation, $F_V'/F_M'$ and qZ showed a linear relationship with a slope of −1.01 aligning almost perfectly with the SV-lake model (4 different experiments with 3 independent biological replicates each time) (Fig. 7). All those observations support the "heterogeneity" over the "economic photoprotection" hypothesis.

## Discussion

To elucidate the nature and mechanism of reversible NPQ in pennate diatoms, we exploited the capability of generating, in *P. tricornutum*, transgenic lines with different Lhcx1 expression and NPQ capabilities

but with similar growth, PSII photochemistry, and light-harvesting pigment composition (Fig. 1). The only significant difference was the distribution of photoprotective xanthophylls normalised to β-carotene pigments, which strongly correlates with Lhcx1 levels (Fig. 2). This dependence suggests potential epistatic control of Lhcx1 over xanthophyll synthesis pathway, similar to the control PsbS exerts over zeaxanthin in *Arabidopsis*[38]. This "molecular titration" approach revealed a robust phenomenological relationship in which NPQ is proportional to [Lhcx1] × DES, valid across nine strains, with very different xanthophyll pool sizes, Lhcx1 concentrations, and NPQ capacities ranging from 0 to roughly 3 times the WT level. It remained valid in two growth conditions despite significant differences in pigment content, $F_V/F_M$ and σPSII (Table 1). The substantial overlap between LL and IL on the NPQ vs DES*[Lhcx1] plot (Fig. 3h) suggests that the quencher *Q* has consistent quenching efficiency across the two growth conditions. We note however that exact comparison of *Q*'s efficiency in quenching Chl *a* fluorescence between LL and IL growth conditions would require expressing the concentration of *Q* per Chl *a* in PSII. This is not accessible here due to the quantification of Lhcx1 concentration via Western blot (it is normalised per the average of D2 and βCF1, see "Methods" section). Nevertheless, this robust phenomenological relationship provides several valuable insights into the fundamental nature of NPQ. First, we can confidently attribute it to a

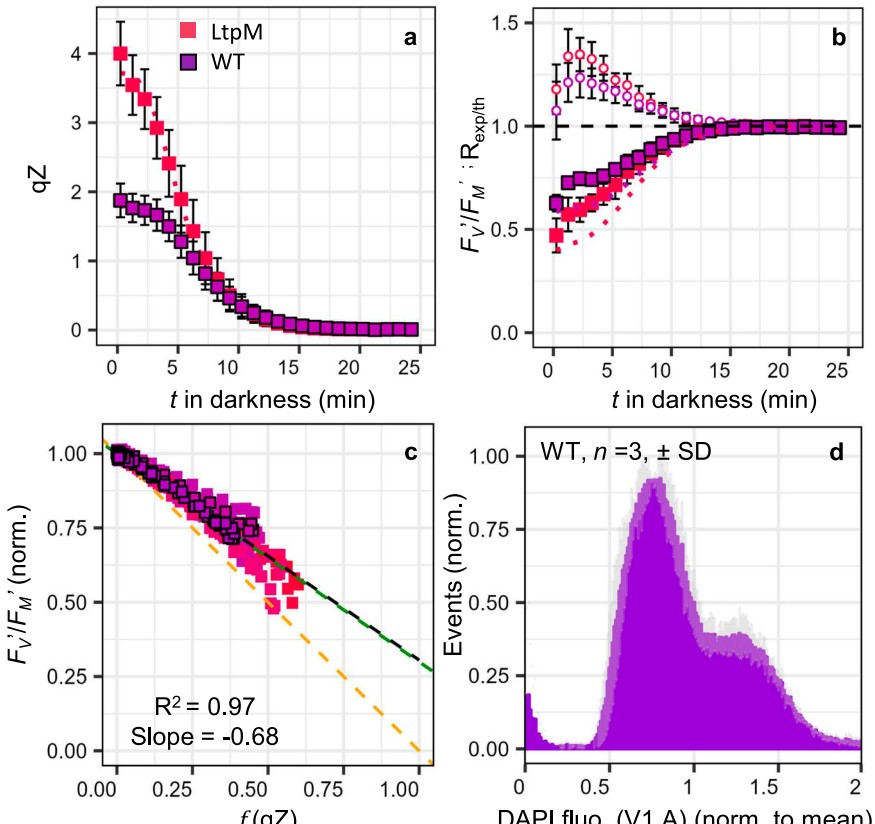

**Fig. 4 | PSII photochemistry vs. NPQ/qZ relationship and the "SV-lake" model.** Relaxation kinetics of the rapidly reversible, diatoxanthin-dependent, qZ component of NPQ (**a**) and of the potential quantum yield of PSII ($F_V'/F_M'$) (normalised to its value at the end of the relaxation) in the dark after induction of qZ under high light (6 min, 750 µmol photons m$^{-2}$ s$^{-1}$). Exemplary curves of mean ± SD of 4 independent biological replicates in *Phaeodactylum tricornutum* wildtype (WT, symbols with black outline) and complemented strain LtpM, grown under moderate light (ML, see "Methods" section). The theoretical $F_V'/F_M'$ according to the "SV-lake" model (dotted line) and the Ratio of experimental-to-theoretical $F_V'/F_M'$ ($R_{exp/the}$, open circles) are shown in (**b**). Relationship between $F_V'/F_M'$ and $f$ (qZ) (relative decrease in $F_V'/F_M'$ due to qZ) during dark relaxation, for all 651 data points corresponding to WT and LtpM (**a, b**) and five other strains (*n* = 4 for WT, LtpG, LtpN,

LtpQ and LtpW, and *n* = 3 for LtpM and LtpY, individually shown in Supplementary Fig. 8). Since Lhcx1 was not quantified under ML, we used the maximum qZ reached to establish a colour code equivalent to that of Fig. 1 for all panels. The black dashed line is the linear regression between $F_V'/F_M'$ and $f$ (qZ) and the theoretical relationship according to the "SV-lake" model (orange dashed line) is shown. Histogram of the distribution of mean cellular DAPI fluorescence (V1. A channel) (proxy of DNA/cell) measured by flow cytometry in the WT (mean ± SD (grey shading), *n* = 3, cytometry results for 5 other mutant strains shown in Supplementary Fig. 10) (**d**). The green dashed line in (**c**) represents the theoretically predicted $F_V'/F_M'$ vs. $f$ (qZ) relationship for a "heterogeneous SV-lake model" situation, compatible with the intercellular heterogeneity measured in (**d**) (see Supplementary Text 2).

single component and reject models proposing multiple quenching sites and mechanisms for NPQ in pennate diatoms, whether it be a direct role of luminal pH, PSII antenna uncoupling or a second quencher. In this work, we rejected the "PSII antenna uncoupling" hypothesis thanks to the observed proportionality between σPSII′ and $F_V'/F_M'$ during NPQ/qZ relaxation (Fig. 5). We propose that variations in σPSII′, stemming from changes in $F_V'/F_M'$, have been misattributed to changes in optical cross-section in previous works. Models involving multiple quenching sites might describe situations where the reversible qZ is accompanied by a slowly relaxing component, such as that related to PSII photodamage (qI) (a difficult bias to control for in fluorescence lifetime experiments[27,32,33]), which we ensured was absent from our protocols. The role of luminal pH in the NPQ of *P. tricornutum* seems limited to regulation of the diadinoxanthin de-epoxidase[48,55] without *direct* pH sensing as seen in green organisms where LHCSR3 and/or PsbS act as pH sensors[21,38,39]. Two recent studies support this distinction by showing that mutation of acidic residues in Lhcx1 –that are involved in pH sensing by LHCSR[56]- does not affect NPQ[45,50]. Our data reinforce this idea. Specifically, if the range of pH values examined here (sufficient to reach DES values from 0 to 60%) influenced NPQ independently of XC, this would disrupt the linear relationship we observe between NPQ and DES, but no such deviation is observed.

Given that qE traditionally refers to the fast-relaxing NPQ *directly* triggered by luminal pH-sensing antenna proteins, we conclude that qE does not accurately represent the fast-relaxing NPQ component in pennate diatoms. Instead, this component aligns more closely with zeaxanthin-dependent quenching (qZ), first described in *Arabidopsis* PsbS mutants[21,49]. Indeed, qZ is an NPQ component displaying slower relaxation than qE, as it strictly follows the conversion of the de-epoxidized pigment (zeaxanthin) back to the epoxidized form (violaxanthin). In plants, qE is also enhanced by zeaxanthin but the kinetics of its induction and relaxation differ from those of the xanthophyll cycle. The strong correlation between fast-relaxing NPQ and DES in our work suggests that this NPQ component is better described by qZ than by qE[48], if the definition of qZ is broadened to "de-epoxidized xanthophyll-dependent quenching".

This robust relationship provides new information to ponder the molecular mechanism of NPQ/qZ and the nature of the SV-quencher. Recent studies using different methods[57–59] argued that lutein acts as the non-photochemical quencher in green organisms' qE by accepting excitation energy transfer from excited Chl *a*. Interestingly, excitation energy transfer to the S$_1$ state of zeaxanthin has also been suggested as the dominant quenching mechanism in *Nannochloropsis oceanica*[60]. While both *Nannochloropsis* and diatoms lack lutein, DT and

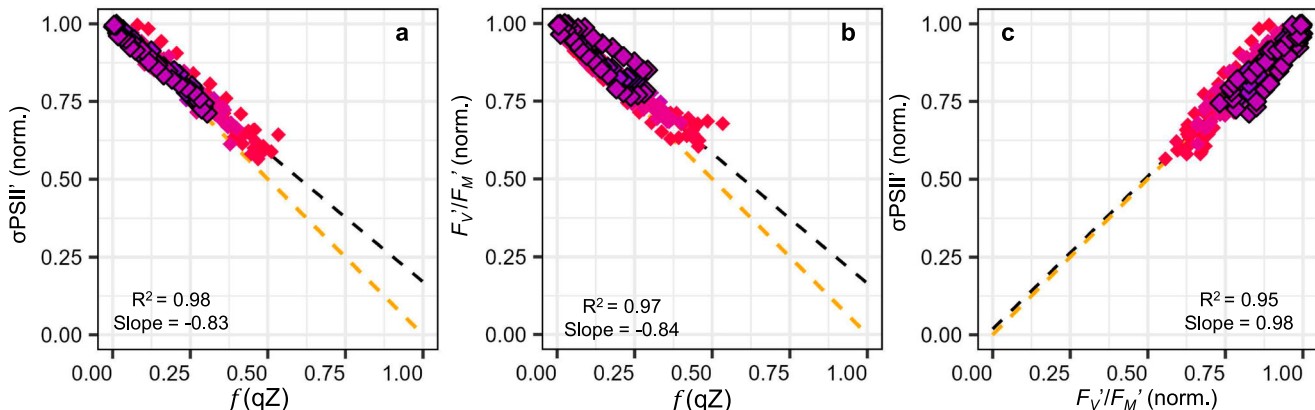

**Fig. 5 | Relationships between NPQ/qZ, PSII photochemistry and functional absorption cross-section are incompatible with the "antenna uncoupling" hypothesis.** The PSII absorption functional cross-section ($\sigma$PSII′) and potential quantum yield for photochemistry ($F_V'/F_M'$) were monitored during relaxation of the rapidly reversible, diatoxanthin-dependent, qZ component of NPQ in the dark and plotted vs. the relative decrease due to qZ ($f$ (qZ)), in *Phaeodactylum tricornutum* wildtype (black outline symbols) and four other Lhcx1-mutant strains (LtpM, LtpN, LtpW, LtpY, see Fig. 1) grown under moderate light (ML) and priorly exposed to 6 min of high light (see "Methods" section). The relationships between $\sigma$PSII′ and $f$ (qZ) (**a**), between $F_V'/F_M'$ and $f$ (qZ) (**b**) and between $\sigma$PSII′ and $F_V'/F_M'$ (**c**) are fitted by linear regressions (black dashed lines) and the theoretical relationships are displayed as orange dashed lines. The data combine 3 independent biological replicates for all strains (except 2 for LtpM), all 281 data points are shown. Since Lhcx1 was not quantified under ML, we used the maximum qZ reached to establish a colour code equivalent to that of Fig. 1 for all panels.

zeaxanthin share an identical terminal ring structure (two in the case of zeaxanthin) to the one putatively involved in lutein's quenching activity. The comparison with *Nannochloropsis*, a closer relative to diatoms, is particularly relevant here because zeaxanthin, like DT for *P. tricornutum*, also requires interaction with an Lhcx protein to allow NPQ[20,60]. This interaction could simply be the binding of xanthophylls on Lhcx1, as supported by the latest biomolecular and structural knowledge of Lhcx1. Indeed, the mutation of a tryptophan residue conserved in all LHCSR and Lhcx proteins (Lhcx4 being a crucial exception) and located in the vicinity of a putative xanthophylls binding pocket, significantly impairs NPQ capacity in *P. tricornutum*[50]. Nevertheless, Lhcx1-DT binding is not the only possible mechanistic model for which DT interacting with Lhcx1 could form an SV-quencher. Like suggested for lutein with CP29 in plants[59], an Lhcx1 binding pocket could allow for more planar s-trans conformer of the xanthophyll, of which the $S_1$ state could form the quencher. Additionally, in *Arabidopsis*, zeaxanthin accumulates at the antenna periphery to induce quenching, rather than being converted from violaxanthin strongly protein-bound within the antenna[61]. The further observation that zeaxanthin-related quenching occurs in isolated membranes[62], but not in isolated complex, suggests that the peripheral zeaxanthin is weakly linked and lost during the purification process[61]. In contrast, in *P. tricornutum*, isolated FCPs contain both DD and DT[40] and are capable of quenching[63].

The "molecular titration" approach used here for Lhcx1 could also be applied to the other isoforms providing NPQ capacity, Lhcx2 and Lhcx3[29], to determine whether the quenching efficiency of DT depends on the isoforms it interacts with. In this model, Lhcx1 would determine the partition between xanthophylls involved in NPQ/qZ and other pools, and from there, the slope of the NPQ vs. DT relationship. A similar idea was initially put forward by Schumann et al.[35], but before discovering Lhcx proteins, this hypothesis could not be convincingly demonstrated since both Lhcx concentrations and the size of xanthophylls pools vary with growth conditions[14]. We can anticipate limits to the simple relationship between NPQ, Lhcx1 and DES: when the XC enzymes are fast compared to the equilibrium time between the different pools of xanthophylls, the DES measured experimentally (on total xanthophylls) could be different from the proportion of DT interacting with Lhcx. We propose that this explains the deviation from proportionality between qZ and DES in some of the strains grown in IL

(Supplementary Fig. 6), consistent with results from refs. 36,64. In some IL-grown strains, some DT remains when NPQ is fully relaxed (Supplementary Fig. 6), likely indicating that the qZ-involved xanthophyll pool undergoes faster epoxidation than the lipid-phase soluble one. This interpretation is also supported by the absence of deviation in LL-grown cells, where qZ relaxation (DT epoxidation) is two-fold slower, allowing more time for DES equilibration between the xanthophyll pools.

The SV-lake model also predicts a unique relationship between the yields of photochemical and non-photochemical processes (Eq. 2) which is perfectly respected in the pennate diatom *Plagiostriata* sp. (Fig. 7) and could be observed in 7 strains of *P. tricornutum* under certain conditions (after acclimation to darkness for 40 h) (Fig. 6). We also show that, without dark acclimation in *P. tricornutum*, the potential photochemical yield and functional cross-section of PSII exceed what is expected in an SV-lake model for a given level of qZ (Figs. 4c, and 5). Similar observations led to alternative models of "economic quenching" (in the case of excess $F_V'/F_M'$[12,27]) or "antenna uncoupling" (in the case of excess $\sigma$PSII[29]). If we excluded the "antenna uncoupling" hypothesis, the discrepancy we observed remains compatible with the "economic quenching" model. However, it is difficult to imagine another process that would compensate for this discrepancy only in some contexts, like in *Plagiostriata* sp. or in *P. tricornutum* after 40 h of dark acclimation, so that the expected relationship in an SV-lake model is respected *by chance*. The most parsimonious interpretation is that the SV-lake model correctly describes the photosynthetic unit of pennate diatoms, but at least one implicit assumption behind Eq. 2 is not satisfied when "excess" photochemistry is measured. One implicit assumption is that properties/parameters of the SV-lake model ([$Q_A$], [$Q$], rate constants) are *homogeneous across all "lake" domains/cells* and *invariant during NPQ relaxation*. We demonstrated that realistic levels of heterogeneity (Supplementary Text 2) between cells/domains obeying the SV-lake model could reproduce the apparent excess of photochemistry measured (Fig. 4c). Furthermore, experimental approaches promoting intercellular homogeneity (such as 40 h incubation in darkness) eliminated this discrepancy (Fig. 6), strongly supporting the "heterogeneity hypothesis". If 40 h incubation in darkness induces closer, but not always perfect, agreement with the SV-lake model (Supplementary Fig. 9), it might be that intracellular heterogeneity remains (e.g.,

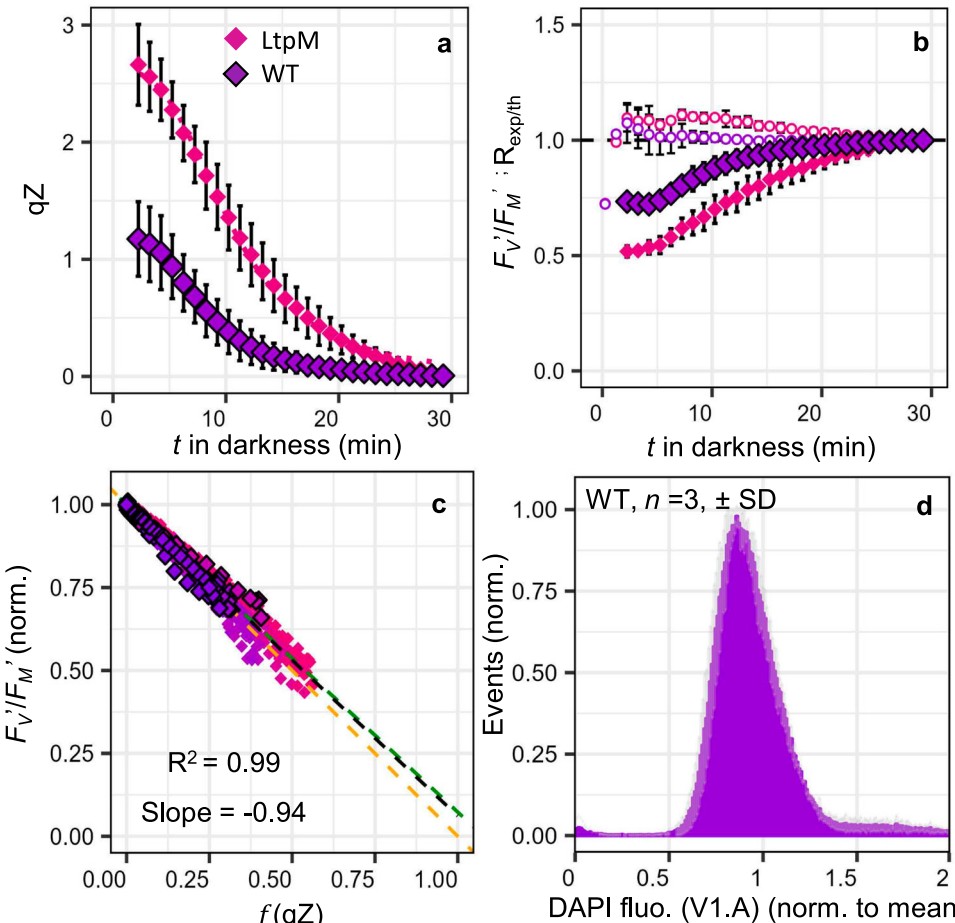

**Fig. 6 | PSII photochemistry vs. NPQ/qZ relationship in homogeneous (synchronised) cell population meets expectations of the "SV-lake" model.**
Relaxation kinetics of the rapidly reversible, diatoxanthin-dependent, qZ component of NPQ (**a**) and of the potential quantum yield of PSII ($F_V'/F_M'$) (normalised to its value at the end of the relaxation) in the dark after induction of qZ under high light (6 min, 750 μmol photons m$^{-2}$ s$^{-1}$). Exemplary curves of mean ± SD of 4 independent biological replicates in *Phaeodactylum tricornutum* wildtype (WT, purple symbols with black outline) and complemented strain LtpM (pink symbols). Strains, grown in moderate light (ML), were incubated in the dark for 40 h to reduce physiological heterogeneity (see text) before measurements. The theoretical $F_V'/F_M'$ according to the "SV-lake" model (dotted line) and the Ratio of experimental-to-theoretical $F_V'/F_M'$ ($R_{exp/th}$, open circles) are shown in (**b**). Relationship between $F_V'/F_M'$ and $f$ (qZ) (relative decrease in $F_V'/F_M'$ due to qZ) during dark relaxation, for all 721 data points corresponding to WT and LtpM (**a**, **b**) and five other strains ($n = 4$,

for WT, LtpM, LtpN, LtpW and LtpY, and $n = 2$ for LtpG and LtpQ, individually shown in Supplementary Fig. 8. Since Lhcx1 was not quantified under ML, we used the maximum qZ reached to establish a colour code equivalent to that of Fig. 1 for all panels. The black dashed line is the linear regression between $F_V'/F_M'$ and $f$ (qZ) (relative decrease in $F_V'/F_M'$ due to qZ) and the theoretical relationship according to the "SV-lake" model (orange dashed line) is shown. Histogram of the distribution of mean cellular DAPI fluorescence (V1.A channel) (proxy of DNA/cell) measured by flow cytometry in WT acclimated to ML and then incubated in the dark for 40 h (mean ± SD (grey shading), $n = 3$, cytometry results for 5 other mutant strains are shown in Supplementary Fig. 10) (**d**). The green dashed line in (**c**) represents the theoretically predicted $F_V'/F_M'$ vs. $f$ (qZ) relationship for a "heterogeneous SV-lake model" situation, compatible with the intercellular heterogeneity measured in (**d**) (see text and Supplementary Text 2).

heterogeneous "PSII-lake" domains with different properties within a cell[51]). The possibility that parameters describing the lake model vary during the NPQ relaxation should however also be considered. Lately, questions have been raised regarding possible variations in the rate constant of PSII intrinsic losses[65]. We also recently observed a decrease in the trans-thylakoidal electric field (Δψ) after a high light-to-dark transition (Croteau et al., forthcoming), which could increase photochemical yield by reducing the probability of charge recombination[66,67]. Excess photochemistry under high NPQ/qZ conditions could also result from a temporarily more oxidized $Q_A$ compared to the situation when qZ has fully relaxed. However, heterogeneity is our favourite hypothesis, a reasonable one that can naturally explain biases in $F_V'/F_M'$ vs. NPQ/qZ relationships (always in the direction of excess photochemistry) without compromising the predictive power of the lake model. It is tempting to propose that the agreement with the lake model relates to the structural simplicity of pennate diatoms thylakoids, which are devoid of the grana stacks/lamella ultrastructure

found in plants[68] or appressed/non-appressed domains found in green algae[69] and display limited PSII segregation[70].

Although its full range of validity remains to be explored, the SV-lake model can be seen as a null hypothesis to test intermediate models between lake and puddle models[23], models including heterogeneity (Supplementary Text 2), additional quenching mechanisms or novel concepts[65]. The NPQ/qZ model presented here for pennate diatoms is remarkably simple compared to models for green organisms[71] or the recent model for *Nannochloropsis*[20] and offers unprecedented predictive power. Considering other characteristics discussed in refs. 1,11, we propose that *P. tricornutum* can be used as a "natural reductionist system" to improve our understanding of photosynthesis regulations. For example, xanthophyll cycle activity can be measured in vivo through chlorophyll fluorescence measurements[48]. We see great potential in generalising the present approach - using a series of transgenic strains differing in the amount of Lhcx1 and xanthophylls, all other factors being equal - to study the orchestration of

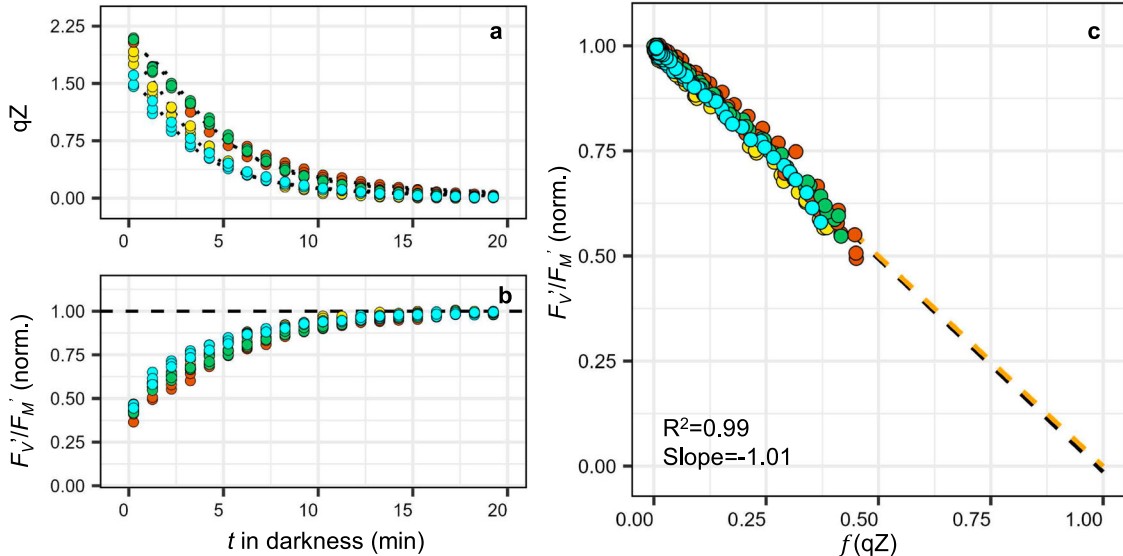

**Fig. 7 | PSII photochemistry vs. NPQ/qZ relationship in the pennate diatom *Plagiostriata* sp. meets expectations of the "SV-lake" model.** Relaxation kinetics of the rapidly reversible, diatoxanthin-dependent, qZ component of NPQ (**a**) and of the potential quantum yield of PSII ($F_V'/F_M'$) (normalised to its value at the end of the relaxation) (**b**) in the dark after induction of qZ under high light (6 min, 750 μmol photons m$^{-2}$ s$^{-1}$) in the pennate diatom *Plagiostriata* sp. acclimated to

moderate light (see "Methods" section). The four colours represent four different experiments conducted at different times on biological triplicates (total $n = 12$). Linear regression (black dashed line) of $F_V'/F_M'$ vs. $f$ (qZ) (relative decrease in $F_V'/F_M'$ due to qZ) on all 240 data points and theoretical relationship according to the "SV-lake" model (orange dashed line) (**c**).

photosynthetic regulations in more detail. Using this "NPQ-dial" as an entry point allows us to extract phenomenological laws describing how NPQ/qZ modulates various other aspects of photosynthesis, such as the redox state of various players in the photosynthetic chain ([Q$_A$] is conveniently assessed through the parameter qL in a lake model only[23]), the partition between cyclic and linear electron flows, or the susceptibility to PSII photodamage. Such an approach has the potential to move diatoms from a footnote to photosynthesis research on green organisms and cyanobacteria, to organisms of predilection to tackle fundamental questions and strengthen our modelling capabilities.

## Methods
### Culture and growth conditions
Wildtype *Phaeodactylum tricornutum* Bohlin CCAP 1052/1A (Pt2) and 13 transgenic Lhcx1 lines were grown in filtered (0.2 μm) seawater with added *f*/2 medium in 40 mL flasks. The transgenic lines consisted of the *Lhcx1* knock-out strain (Ko6) and 12 complemented lines expressing different levels of *Lhcx1* in the Ko6 background, first generated in ref. 45. The complemented strains are called LtpA, LtpB, etc. for *Lhcx1* Talen target site modified plasmid strain A, B, C, etc. Cells were grown at 19 °C under three different light conditions: low light (LL) (5 μmol photons m$^{-2}$ s$^{-1}$) or medium light (ML) (40 μmol photons m$^{-2}$ s$^{-1}$) both under a 12 L:12D photoperiod, and intermittent light (IL) (cycles of 55 min dark and 5 min at 40 μmol photons m$^{-2}$ s$^{-1}$)[12]. Cultures were monitored every 2-to-3-days for cell concentration using a Multisizer 3 Coulter counter (Beckman Coulter, CA, USA). For experiments, they were sampled in the exponential phase (between 0.7 and 1.5 × 10$^6$ mL$^{-1}$, always 4-to-5 h into the light phase of the photoperiod), concentrated (10–15 times) by centrifugation (5000 RPM, 6 min), then left to recover on an agitator for 30 min under very low light (approximately 1 μmol photons m$^{-2}$ s$^{-1}$). Cultures were rediluted approximately 10 times when reaching 2 × 10$^6$ cells/mL to ensure they remained in the exponential phase. Growth rates were calculated as the slope of the ln-transformed concentration versus time (in days) in the exponential phase.

### Western blotting and Lhcx1 protein quantification
Lhcx1 immunodetection on IL cultures was performed as previously done for LL cultures in (Supplementary Fig. 14E in ref. 45), using a rabbit polyclonal anti-LHCSR antibody from *Chlamydomonas* (gifted from K. K. Niyogi, University of California, Berkeley, USA, dilution 1: 5000) which can detect all of *P. tricornutum* Lhcx isoforms[33,42,46]. In this study, two proteins, instead of one, were used as loading control, the ATP synthase-βCF$_1$ and the PSII-D2 subunits (detected thanks to the antibodies gifted by J-D. Rochaix, University of Geneva, Geneva, Switzerland, at a 1: 10,000 dilution each). The different protein bands were quantified by Image Lab v5.0 software (Bio-Rad) and values derived by the average of the βCF$_1$ and D2 signals were used to calculate the Lhcx1 concentrations. This allowed comparisons of Lhcx1 concentrations between the two different growth conditions, LL and IL, and between wildtype and transgenic lines expressing different Lhcx1 content. Wildtype and the highest Lhcx1 expressor, LtpM, were blotted together on the same membrane (twice for each strain giving four calibration curves) (Fig. 1 and Supplementary Fig. 1) for ranges of total protein loadings (between 1 and 10 μg). From the linear fitting of the Lhcx1 versus total protein loading obtained in LL and IL, we calculated a calibration factor of 2.49 ± 0.62 to account for the increase in Lhcx1 concentration from LL to IL. We normalised all LL Lhcx1 values from ref. 45 between 0 and 1 and divided these values by the 2.49 calibration factor so that LL and IL [Lhcx1] were then comparable on the same relative scale.

### Pigment analysis
For pigment analysis, cells were concentrated 10–20 times by centrifugation and then let for 30 min under very low light (approximately 1 μmol photons m$^{-2}$ s$^{-1}$) in 15 mL Falcon tube under gentle agitation. For dark-adapted cells, the sampling was done directly in the Falcon. For light-acclimated cells, sampling was done directly in the plateholder in the fluorometer when steady-state NPQ was reached or during NPQ relaxation in darkness. 50 μL was sampled and immediately dropped in 950 μL of pure methanol, quickly vortexed (2 s) and flash-frozen in liquid nitrogen before storage at −80 °C. Pigment analysis was done following[72] protocol, with a Shimadzu Prominence- I LC-2030C 3D

HPLC (Shimadzu Corporation, Kyoto, Japan) equipped with a Waters Nova-Pak C18 4 μm 3.9 × 150 mm column (Waters Corporation, Milford, USA). In our hands, Chl $a$ degradation is inevitable even on a 4 °C refrigerated plate, so we thawed and ran one sample at the time.

**Fluorescence measurements: $F_V/F_M$, σPSII and NPQ calculation**
For most fluorescence measurements, we used a fluorescence imaging system (SpeedZen, JbeamBio, France), allowing to measure multiple 60 μL samples in parallel, with a green actinic light (532 nm). Minimum ($F_O$) and maximal ($F_M$) fluorescence were measured on dark-adapted samples, before and after a saturating pulse of 250 ms (5000 μmol photons m$^{-2}$ s$^{-1}$, 532 nm), respectively, to calculate the maximal quantum yield of PSII $F_V/F_M = (F_M - F_O)/F_M$. When the minimal ($F_O'$) and maximal ($F_M'$) fluorescence were measured in the dark during NPQ relaxation, following a high light treatment, the potential quantum yield of PSII during NPQ relaxation was calculated as $F_V'/F_M' = (F_M' - F_O')/F_M'$. The maximum quantum yield at the end of the NPQ relaxation was calculated as $F_VR/F_MR = (F_MR - F_OR)/F_MR$ where $F_OR$ and $F_MR$ are the minimal and maximal fluorescence measured when NPQ is fully relaxed.

For NPQ measurements, we distinguished two situations depending on whether the protocol allowed to discern specifically the non-photochemical quenching associated with the rapidly relaxing NPQ component qZ, or the sum of all NPQ components had to be considered (including the slowly relaxing component that we attribute to qI when NPQ relaxation could not be monitored). The qZ component we investigate in this paper is rapidly deployed (3 min under light) or relaxed (approximately 15 min in darkness), and fully relaxed in all treatments for which it was measured (Supplementary Figs. 5 and 8).

To measure NPQ under steady-state illumination, the samples were exposed to seven increasing light-steps (75–750 μmol photons m$^{-2}$ s$^{-1}$) for 3 min (sufficient to reach a steady-state) before measuring the light-acclimated $F_M'$ at each light-step. We then calculated NPQ as $1 - F_M/F_M'$. From these fluorescence values, the relationship between NPQ and light intensity ($E$) was fitted as in ref. 47: NPQ = NPQ$_M \times E^n$/($E50$NPQ$^n + E^n$), where NPQ$_M$ is the extrapolated maximal NPQ value, $E50$NPQ is the light intensity for NPQ half-saturation, and $n$ is the sigmoidicity coefficient. For NPQ vs. pigment analysis, performed on 6 LL and 9 IL strains, this experiment was conducted by a batch of five separated sub-samples which were each collected for further pigment analysis, whether it is at the end of different light-steps or at different times of relaxation in darkness following light stress (450 μmol photons m$^{-2}$ s$^{-1}$) of 6 min (see Fig. 3).

For the NPQ relaxation analysis, we first applied 6 min of light stress (750 μmol photons m$^{-2}$ s$^{-1}$) to induce NPQ and then followed its relaxation in darkness, during 15–30 min depending on growth conditions. The monitoring of $F_O'$ and $F_M'$ began 15 s after the transition to darkness and was measured every min afterwards until full relaxation, from which we computed $F_V'/F_M'$, $F_VR/F_MR$. To test the validity of Eq. 2, we reasoned that the influence of slowly relaxing phenomena should be eliminated. Therefore, we used the situation at the end of the dark relaxation, instead of the dark-acclimated situation before the light stress, as a reference for maximal fluorescence ($F_MR$) and maximum quantum yield ($F_VR/F_MR$). This allowed us to calculate the reversible part of NPQ as qZ = $1 - F_MR/F_M'$, and to test Eq. 2 which was adapted from ref. 73 and rewritten as:

$$\frac{F_V'/F_M'}{(F_{VR}/F_{MR})} = 1 - f(qZ) \text{ where } f(qZ) = \frac{(1 - F_{VR}/F_{MR}) \times qZ}{(1 + (1 - F_{VR}/F_{MR}) \times qZ)} \quad (3)$$

The experimental data for time in darkness ≥ 1.25 min was used to test the validity of Eq. 3. The ratio between experimental and

theoretical $F_V'/F_M'$ was calculated as:

$$R \exp / th = \frac{F_V'/F_M'}{F_{VR}/F_{MR} \times (1 - f(qZ))} \quad (4)$$

For measurements of the functional absorption cross-section of PSII (σPSII), we used a Fluorescence Induction and Relaxation (mini-FIRe) fluorometer built by Max Gorbunov (Rutgers University, NJ, USA). In this setup, 3 mL of a sample were left to relax under low light as described above but without centrifugation. The sample was then transferred in a glass tube, then a single turnover flash of intense blue light (100 μs, 455 nm, 60 nm bandwidth) was provided in parallel $F_V/F_M$ or $F_V'/F_M'$ measurements, as well as σPSII (dark acclimated before light treatment) or σPSII' (during NPQ relaxation in darkness)[26]. The optical cross-section of PSII was calculated as σPSII/($F_V/F_M$)[74].

**Flow cytometry**
For six strains in ML, we collected two 15 mL samples. One was directly centrifuged (2500 rpm, 15 min, 4 °C), while the other was incubated in a vented flask in darkness for 40 h before centrifugation. After centrifugation, cell pellets were fixed in cold 70% EtOH, and stored 48 h in darkness at 4 °C. Prior to cytometry, cells were washed once in cold EtOH and once in PBS, then stained by diamidino-2-phenylindole (DAPI, final concentration of 0.5 ng/mL). After 45 min incubation in darkness at room temperature, cells were washed with PBS again. Cytometric analysis was carried out with a MACSQuant Analyser flow cytometer (Miltenyi Biotec, Germany), at least 50,000 events were recorded per sample. The V1.A channel (408 nm excitation, 450/50 nm detection) was used to detect DAPI fluorescence, a proxy for cellular DNA content and forward scatter (FSC) was used to estimate biovolume. Cells in phase G1 are expected to show lower DAPI fluorescence and smaller biovolume compared to cells in phase S, G2 or M[52].

**Curve fitting and statistical analysis**
For statistical analysis, we were interested in trends through a [Lhcx1] dimension (continuous relative scale from 0 to 1) rather than in one-to-one inter-strain comparisons. All parameters were not measured in all strains and growth conditions, but we tried to select the most relevant [Lhcx1] for a given question. Therefore, the sample size was not predetermined nor randomised. To test for the significant influence of [Lhcx1] on a given parameter, we used linear regressions calculated over the mean value of the parameter in each strain/growth condition (for homogeneous weights between different [Lhcx1] coordinates when the number of biological replicates is heterogeneous between strains). All relationships are shown in Table 1 and all mean values ± SD are in Supplementary Data 1. We then compared the distribution of mean values of parameters among strains between growth conditions (LL and IL) (independent factors) to confirm that we were studying the effects of NPQ in two PSII-antenna-systems with significantly different features. We selected the dependent variables that did not directly include Lhcx1 or NPQ: $F_V/F_M$, σPSII, optical absorption cross-section of PSII, and concentration of different pigments (Table 1). To access significant mean differences between growth conditions for these variables we used a type III one-way Analysis of Variance (ANOVA) with Satterthwaite's method (R software, package *lmerTest*), to consider an uneven number of strains used and the varying number of biological replicates between growth conditions and independent variable measured (Supplementary Data 1).

**Reporting summary**
Further information on research design is available in the Nature Portfolio Reporting Summary linked to this article.

## Data availability

Additional data for all replicates can be found in Supplementary Data 1. The source data generated in this study have been deposited in the figshare database [https://doi.org/10.6084/m9.figshare.28233944].

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

## Acknowledgements

We thank Francis-André Wollman for critical reading of the manuscript, as well as Johann Lavaud, Bernard Lepetit and Bernard Genty for fruitful discussions. We also thank K. K. Niyogi and J-D. Rochaix for the gifted antibodies and Chiara Giossi and Marcelo Orlando for help with the measurement of Chl *a* standard. D.C. and B.B. acknowledge the support

of the European Research Council (ERC) PhotoPHYTOMIX project (grant agreement No. 715579). D.C., B.B. and A.F. also acknowledge funding from the BrownCut Projet (ANR-19-CE20-0020). A.F. acknowledges funding from Fondation Bettencourt-Schueller (Coups d'élan pour la recherche francaise-2018), the "Initiative d'Excellence" program (Grant "DYNAMO," ANR-11-LABX-0011-01) and by the EMBRC-FR-"Investisse-ments d'avenir" program (ANR-10-INBS-02).

## Author contributions

D.C. and B.B. designed the study. D.C. performed all biophysics measurements and D.C. and B.B. analysed the data. M.J. and D.C. performed the Western blot analysis and analysed the results with support from A.F. The original draft of the manuscript was written by D.C. All the authors discussed the results and commented on the manuscript.

## Competing interests

The authors declare no competing interests.
