## [Transparent Peer Review file · Nature Communications]

Pennate diatoms make Non Photochemical Quenching as simple as possible, but not simpler.

Corresponding Author: Dr Benjamin Bailleul

Version 0:

Reviewer comments:

Reviewer #1

(Remarks to the Author)

Good things first. The authors have studied *P. tricornutum* lines that express different levels of Lhcx1 proteins, and investigated the relationship between Lhcx1 abundance and NPQ/qZ amplitude. By using two non-stressful light conditions and modulating diatoxanthin levels in these lines they showed that NPQ is proportional to the abundance of Lhcx1 and the relative content of diatoxanthin. They conclude that the interaction between Lhcx1 and diatoxanthin creates a homogeneous S-V quencher responsible for NPQ in this strain (which was also confirmed in the pennate diatom *Plagiosiriata* sp, see last figure). Therefore, the authors draw the following conclusions (1) this analysis confirms the validity of the S-V lake model in diatoms; (2) the robust relationship between NPQ, [Lhcx1] and DES supports the idea that a single component dominates the rapidly-reversible quenching response in diatoms.

Their major finding of this research is that the non-photochemical quenching model presented here for pennate diatoms is remarkably simpler than previous models for other species of the green lineage, and offers great predictive power. *P. tricornutum* is therefore a 'natural reductionist system' of great utility in understanding the regulatory mechanisms of light-harvesting.

Nevertheless, there are some points that need further improvements before the manuscript can be accepted for publication. They are listed below:

- It is not clear why the authors use the numerical values of NPQ to indicate the linear correlation between NPQ and DT in the first three figures and then base the entire subsequent analysis on the qZ values. Does this mean qE is excluded from the analysis? The authors state that this component of NPQ "is analogous to the strictly zeaxanthin-dependent and slowly relaxing component called qZ observed in *Arabidopsis* PsbS mutant – ref. 25 and 51". I think more data on the actual analogy between the mechanisms of the two organisms should be provided.
- A key molecular effector for NPQ is Lhcx, and in *P. tricornutum* isoforms 1, 2 and 3 influence the amplitude of NPQ (see ref. 33). Have the authors investigated whether the expression level of the other isoforms changes under the conditions and genotypes analyzed? This could affect their main conclusions.
- The reference list needs deep revision; for example, ref nr 3 and nr 34 need the text reviewed. Some references include the DOI, others do not. Reference nr 48 is a 2022 publication. Line 323: "...potential epistatic control of Lhcx1 over xanthophyll synthesis pathway, similar to the control PsbS exerts over zeaxanthin in *Arabidopsis*," while ref. nr 56 says other, I believe.
- The quantification of Lhcx in the LL samples was not conducted by the authors, but values from a previous publication (ref. 48) were used. I don't think this approach is rigorous, especially considering that the culture conditions in the previous publication do not match those reported in the methods of the present manuscript.
- Line 362, could the sigmoidal-like relationship between NPQ/qZ and DES, during the rapid relaxation of both NPQ and qZ, be an index of a more complex NPQ model, and suggest the coexistence of other quenching mechanisms, possibly related to other components such as lumen acidification or transthylakoid pmf? Components that, when considering only the kinetics of qZ, might be underestimated? This might affect the main conclusion of the manuscript, i.e. *P. tricornutum* as a natural reductionist system.

Minor points:

- I think the color code in the figures could also be accompanied by the name of the line used in that panel. Having the genotypes immediately evident in the panel, rather than searching through long captions, simplifies reading and understanding.
- Figure S1: why is there so much diversity in the optical densities of the D2 bands of different replicates?
- Line 170, (Fig. 1d-g)
- In the methods, it is not indicated how the total proteins content of the samples used for the western blot were determined.

Reviewer #2

(Remarks to the Author)

The work from Croteau et al. aims at investigating the NPQ mechanisms in the diatom *Phaeodactylum tricornutum*, using an interesting quantitative approach to distinguish between different hypotheses.

The motivation of the work is also to expand the use of an eukaryotic algae, not belonging to the green lineage. This is highly relevant for the field, as it is becoming evident that regulation of photosynthesis mechanisms is different in diverse organisms and that results obtained in models such as *Arabidopsis* and *Chlamydomonas* cannot be generalized, differently from what is often done in the literature.

The work presents a solid and data-based approach to analyse NPQ of these organisms and is a valuable contribution to the field.

Comments:

1. Lines 108 and the following from the Introduction. This part is a bit hard to follow, especially for non-experts. In a journal targeted to a wide audience, an effort to explain better some concepts from the field should be taken. e.g., lines 112 the concept of 'economic quenching' in plants should be better described. Since this is also in the title of the present manuscript, it should not be left to a reference to previous literature.
2. Lines 125 'Another argument against the original idea that DT is the SV-quencher is the strict requirement of a second molecular effector for NPQ: Lhc proteins 45,46, which are LHC stress related sub-family ...'. I find this part is a bit confusing. Assuming that a binding of DT to LHCX is the more parsimonious hypothesis to explain the data. However, DT bound to LHCX could still be the quencher; only it would require the protein binding.
3. Line 218. One of the conclusions is to exclude that there is a 'direct pH regulation of LHCSR3/PsbS as in green organisms 42,43'. I recommend some caution with this conclusion. Although LHCX alone is not inducing quenching and clearly DT is essential for NPQ activation, it should be remembered that activation of the xanthophyll cycle also depends on lumenal pH. The possibility that quenching requires low pH activation of LHCX1 cannot be excluded. This is a possibility, assuming that LHCX1 is activated at the same time or after DT synthesis. Since VDE in diatoms is activated at less acid pH than in plants, according to some old literature, this is a realistic hypothesis.

Minor point

Line 48 "(PS) have similar absorption spectra and do not form supercomplexes (see forthcoming review (Croteau et al.,', the citation is missing

Reviewer #3

(Remarks to the Author)

The authors have provided a very thorough and useful analysis of the LHCX/DT role in NPQ in diatoms, from which the field will surely benefit.

I have a single major point that required discussion/revisions from the authors prior resubmission, however.

Both in the abstract, text and discussion, the authors mention that there is likely binding of DT to LHCX (as for example stated in the abstract). This is not however based on any structural evidence, isolation of LHCSR or even reconstitution with pigments, neither in this work nor in the past, to my understanding. For example, in plants it was shown that zeaxanthin does not bind to any LHC, even if pigment binding sites might be available from a structural point of view (Xu et al *Scientific Reports* 2015).

All these points should be rediscussed, at the moment I had the impression that it seems that from this work we could conclude that there is binding which instead we cannot conclude based on the presented data. If this point was crucial for some of their analysis, the authors should explain whether their results/conclusions will hold also in case DT does not bind to LHCX. If instead this is only a speculation, it should be more carefully rephrased in all the points given the lack of evidence.

Version 1:

Reviewer comments:

Reviewer #1

(Remarks to the Author)

The authors have satisfactorily addressed my previous questions and concerns. The manuscript has been significantly improved during the revision process. I have no further questions and recommend the publication of the work in Nature Communications.

Luca Dall'Osto

Reviewer #2

(Remarks to the Author)

In my opinion, authors answered satisfactorily to all reviewers' comments.

I'd only add a minor point on the response to reviewer 1 on NPQ components nomenclature, since as stated by authors, can be confusing.

In plants the distinction between qE and qZ is made based on the kinetics, with qE being the fastest component. In motivating their choice to call the NPQ component qZ , I would recommend making a direct comparison of the diatoms and plants NPQ relaxation kinetics. I tried to reconstruct this comparison looking into the supplementary material and it was not straightforward. In this discussion it should also be considered that in plants qE is also dependent on zeaxanthin. It is not necessary for qE but its presence enhance it.

I also realized that in the whole supplementary material there is not an example of a NPQ measurement. I understand this is redundant and published many times, but since the manuscript focuses on these measurements, a few examples in supplementary material could still be useful for some readers.

Reviewer #3

(Remarks to the Author)

The authors addressed my comments and I also appreciate their thorough discussion of all other reviewers' comments.

**REVIEWER COMMENTS**

Before addressing the different comments and questions of the three reviewers, we made two changes
which were not requested by the reviewers. Since receiving the reviews, the first author of this paper,
Dany Croteau, successfully defended his PhD. During this process, the comments of members of his jury
encouraged us to make two additional modifications.

First, a jury member reminded us that the efficiency of a molecule in quenching the Chl *a* fluorescence
from PSII can only be estimated through the ratio between NPQ and the concentration of this molecule
*normalized to Chl a in PSII*. This could not be done in our work (because in Western blots Lhcx1
concentration was normalized instead to D2+ β CF1 - DES is dimensionless). Although this is a minor
aspect of our work, we felt it was worth acknowledging and have added a sentence in the discussion to
address this point:

- L357-360: “We note however that exact comparison of *Q*'s efficiency in quenching Chl *a*
fluorescence between LL and IL growth conditions would require expressing the concentration of
*Q* per Chl *a* in PSII. This is not accessible here due to the quantification of Lhcx1 concentration
via Western blot (it is normalized per (D2 + β CF1), see Methods).”

Another jury member, while agreeing with our conclusion that the pigment composition is consistent
across strains except for the xanthophyll pool, raised a concern about the apparent overestimation of
chlorophyll (Chl) *a* concentration in our data. This member noted that the concentrations of all pigments
appeared low compared to the literature on *Phaeodactylum tricornutum* but *only when normalized per Chl*
*a*. He suspected that this gap came from the Chl *a* standard used in our HPLC quantifications. During the
reassessment of our standards, we confirmed that calibration for the quantification of Chl *a* was indeed
incorrect of a factor ~ 0.7 . We have since corrected all pigment data, which leads to pigment
quantifications which are systematically 42% higher than those reported in the previous manuscript, when
normalized per Chl *a*. This correction does not affect the text, because the absolute values have not been
discussed in the manuscript. The conclusions based on the relative changes between the strains are not
affected. We added this member, Bernard Lepetit, as well as two Ph. D. students who helped with this,
Chiara Giossi and Marcelo Orlando, in the Acknowledgments.

**Reviewer #1 (Remarks to the Author):**

Good things first. The authors have studied *P. tricornutum* lines that express different levels of Lhcx1
proteins, and investigated the relationship between Lhcx1 abundance and NPQ/qZ amplitude. By using
two non-stressful light conditions and modulating diatoxanthin levels in these lines they showed that NPQ
is proportional to the abundance of Lhcx1 and the relative content of diatoxanthin. They conclude that the
interaction between Lhcx1 and diatoxanthin creates a homogeneous S-V quencher responsible for NPQ in
this strain (which was also confirmed in the pennate diatom *Plagiosiriata* sp, see last figure). Therefore,
the authors draw the following conclusions (1) this analysis confirms the validity of the S-V lake model in
diatoms; (2) the robust relationship between NPQ, [Lhcx1] and DES supports the idea that a single
component dominates the rapidly-reversible quenching response in diatoms.

Their major finding of this research is that the non-photochemical quenching model presented here for
pennate diatoms is remarkably simpler than previous models for other species of the green lineage, and
offers great predictive power. *P. tricornutum* is therefore a 'natural reductionist system' of great utility in
understanding the regulatory mechanisms of light-harvesting.

>> Thanks to the reviewer for this positive assessment.

Nevertheless, there are some points that need further improvements before the manuscript can be
accepted for publication. They are listed below:

- It is not clear why the authors use the numerical values of NPQ to indicate the linear correlation between
NPQ and DT in the first three figures and then base the entire subsequent analysis on the qZ values. Does
this mean qE is excluded from the analysis? The authors state that this component of NPQ “is analogous
to the strictly zeaxanthin-dependent and slowly relaxing component called qZ observed in Arabidopsis
PsbS mutant – ref. 25 and 51”. I think more data on the actual analogy between the mechanisms of the
two organisms should be provided.

>> There are actually two distinct questions here, each highlighting aspects of the manuscript that could
be clarified further. We appreciate the reviewer for bringing this to our attention!

1- The first question concerns why we refer to NPQ in the initial part of the manuscript but switch to
qZ later on, which may indeed cause some confusion. To clarify, NPQ technically encompasses
multiple components that can be distinguished by examining NPQ relaxation. In *Phaeodactylum*
*tricornutum*, NPQ includes a fast-relaxing component dependent on the xanthophyll cycle
(regardless of whether we attribute this mechanism to qE or qZ, as discussed below) and a slow-
relaxing component associated with photoinhibition-related quenching, or qI. When NPQ
relaxation cannot be tracked—such as when cells are exposed to different light intensities to
generate NPQ vs. intensity curves, or when cells are sampled under steady-state light for pigment
analysis (Fig. 2, 3)—it’s not possible to separate the fast-relaxing component from qI within
NPQ. Therefore, we use “NPQ” in these cases. In contrast, when NPQ relaxation is monitored
and the maximum recovered fluorescence after relaxation can be measured, it becomes possible
to distinguish between the fast- and slow-relaxing components. In such cases, we calculate the
extent of the fast-relaxing NPQ component (qE or qZ) using the relaxed F_M' as a reference. This
procedure is detailed in the "Fluorescence measurements" section of the Methods. We have now
added further clarification in the Results section to make this approach more explicit.

L206-209: “When NPQ relaxation was followed, we observed that the fully relaxed F_M' returns to
a value very close to the initial dark-adapted F_M (Fig. S4a-b). This confirmed that only a fast-
relaxing component of NPQ was significant in our experiments, allowing us to neglect
contributions from photoinhibition-related quenching (qI) in our interpretation of the NPQ data.”

L258-265: “The final part of this work focuses specifically on the relaxation of the NPQ in the
dark, which allows us to mathematically separate the fast-relaxing component of NPQ from the
slow-relaxing one. We achieved this by using, as a reference, the F_M' value at the end of the fast
relaxation phase (see Methods), which was not possible in the first part of this work relying partly
on steady-state NPQ values. To clarify that any minor contributions from slow-relaxing
components of the NPQ, such as qI, is now excluded, we will henceforth refer to the fast-relaxing
component as qZ rather than the more general and less precise term NPQ.”

2. The next question relates to nomenclature. The distinction between qE and qZ is indeed a key
focus of our paper and may lead to some confusion, given existing literature. We appreciate
Reviewer 1 and Reviewer 2 for identifying this point needing clarity in the manuscript. The
established nomenclature (qE, qZ, qT, qI, qH), which effectively describes NPQ in plants, often
falls short when addressing the diversity within microalgae. When naming a new NPQ
mechanism in diverse organisms, we can either assign a new label (e.g., qY or qX—though we

may run out of letters!) or adapt the plant-based nomenclature to reflect biodiversity. There is no
perfect solution, but we have chosen the latter approach. The single fast-relaxing NPQ component
in pennate diatoms is proportional to the amount of de-epoxidized xanthophyll both at steady
state and during relaxation, without including very rapid relaxation related to luminal protonation
of PSII subunits. These features align it more closely with qZ than qE, even though in the initial
description of qZ (and its name: zeaxanthin-dependent quenching) the de-epoxidized pigment is
zeaxanthin in plants (and not diatoxanthin interacting with Lhcx1, as we propose here).
Therefore, extending the definition of qZ to encompass NPQ components strictly dependent on
de-epoxidized xanthophylls (whether zeaxanthin, diatoxanthin, or others yet to be identified) may
be simpler for the NPQ research community than continually introducing new labels like qY or
qX, which could become confusing. We have clarified this reasoning in one short section of the
Results and one long section in Discussion.

L243-245: “This NPQ component being strictly proportional to the proportion of de-epoxidized
xanthophyll, it is analogous to the strictly zeaxanthin-dependent and slowly relaxing component
qZ, observed in *Arabidopsis* PsbS mutants (Nilkens et al., 2010, Kress & Jahns, 2017) (see
Discussion).”

L371-386: “The role of luminal pH in the NPQ of *P. tricornutum* seems limited to regulation of
the diadinoxanthin de-epoxidase (Blommaert et al., 2021, Grouneva et al., 2008) without *direct*
pH sensing as seen in green organisms where LHCSR3 and/or PsbS act as pH sensors (Nilkens et
al., 2010, Li et al., 2000, Peers et al., 2009). Two recent studies support this distinction by
showing that mutation of acidic residues in Lhcx1 –that are involved in pH sensing by LHCSR
(Liguori et al., 2013)- does not affect NPQ (Giovagnetti et al., 2022, Buck et al., 2021). Our data
reinforce this idea. Specifically, if the range of pH values examined here (sufficient to reach DES
values from 0 to 60%) influenced NPQ independently of XC, this would disrupt the linear
relationship we observe between NPQ and DES, but no such deviation is observed. Given that qE
traditionally refers to the fast-relaxing NPQ *directly* triggered by luminal pH-sensing antenna
proteins, we conclude that qE does not accurately represent the fast-relaxing NPQ component in
pennate diatoms. Instead, this component aligns more closely with zeaxanthin-dependent
quenching (qZ), first described in *Arabidopsis* PsbS mutants (Nilkens et al., 2010, Kress & Jahns,
2017) . Indeed, qZ is an NPQ component displaying slower relaxation than qE, as it depends on
the conversion of the de-epoxidized pigment (zeaxanthin) back to the epoxidized form
(violaxanthin). The strong correlation between fast-relaxing NPQ and DES in our work suggests
that this NPQ component is better described by qZ than by qE (Blommaert et al., 2021), if the
definition of qZ is broadened to “de-epoxidized xanthophyll-dependent quenching”.”

- The quantification of Lhcx in the LL samples was not conducted by the authors, but values from a
previous publication (ref. 48) were used. I don’t think this approach is rigorous, especially considering
that the culture conditions in the previous publication do not match those reported in the methods of the
present manuscript.

- A key molecular effector for NPQ is Lhcx, and in *P. tricornutum* isoforms 1, 2 and 3 influence the
amplitude of NPQ (see ref. 33). Have the authors investigated whether the expression level of the other
isoforms changes under the conditions and genotypes analyzed? This could affect their main conclusions.

(We reordered the comments of **Reviewer 1** for a more linear reply):

>> We agree with the reviewer that our approach would lack rigor if we were attempting to replicate the
growth conditions from a separate lab as described in the referenced publication. Given the well-known

challenges in reproducing exact lab conditions across different facilities, taking the quantifications from
Giovagnetti et al. at face value could indeed be overly optimistic. However, in this case, no reproduction
was necessary: we used the exact same growth conditions, as the strains discussed here were initially
generated, cultivated, and analyzed for Lhcx isoforms in our laboratory.

Reference 48 (#45 in the revised manuscript) (Giovagnetti et al., 2022) was produced as part of a
collaboration between our lab and Alexander Ruban's lab. This publication shares two co-authors with
the present manuscript, Marianne Jaubert and Angela Falciatore, who were responsible for generating the
comprehensive Lhcx1-complemented mutant collection and quantifying Lhcx1 levels via Western blots
(see Fig. 5 in Giovagnetti et al., 2022). For this portion of the study, the mutants were grown in our lab's
growth chamber under the exact conditions labeled LL in this manuscript ($5 \mu\text{mol photons m}^{-2} \text{ s}^{-1}$, 12:12,
19°C). The strains remained in the same location and growth chamber since the publication of
Giovagnetti et al. 2022: the Lhcx1 quantification under LL was conducted in late 2020, and the
experiments for this manuscript began in early 2021.

To confirm consistency, we performed additional Western blot analysis for 14 strains under IL conditions,
including two strains (Pt2 and the highest Lhcx1 overexpressor, LtpM) cultivated under LL. The relative
Lhcx1 levels in these two strains (see Fig. 1 and S1) aligned well with those in Giovagnetti et al., 2022.
Based on the reviewer's comment, we now realize this critical detail was insufficiently highlighted in the
initial manuscript and have modified the following sentence to clarify:

L177-179: "We first measured Lhcx1 accumulation in the 14 strains grown under intermittent light (IL)
conditions using Western blots, as we had previously done for these strains grown under the low light
(LL) conditions (first reported in (Giovagnetti et al., 2022))."

L484-486: "The transgenic lines consisted of the Lhcx1 knock-out strain (Ko6) and 12 complemented
lines expressing different levels of Lhcx1 in the Ko6 background, first generated in (Giovagnetti et al.,
2022)."

>> Regarding the detection of other Lhcx isoforms, the LHCSR antibody used in this study (provided by
159 K.K. Niyogi, University of California, Berkeley, USA) is able to recognize all Lhcx isoforms of *P.*
*tricornutum*, as demonstrated in previous studies (Bailleul et al., 2010, PNAS; Taddei et al., 2016, J Exp
Bot; and Taddei et al., 2018, Plant Phy), which we now clarify at L501-504. Therefore, the presence of a
single band in the Western blots indicates that there was no significant accumulation of other Lhcx
isoforms under our experimental conditions (which were chosen specifically to limit this occurrence). In
some IL replicates, a second very faint band appeared, attributed to Lhcx2 and/or Lhcx3 as in (Buck et al.,
2019 and 2021), but it was minimal compared to the dominant Lhcx1 band. We address the issue of
isoform detection in more detail in in the Results and Methods sections to make this clear:

L176-177: "Other isoforms are not significantly expressed in the absence of high light periods (Bailleul *et*
*al.*, 2010) or nutrient deprivation (Taddei et al., 2016)."

L501-504: "Lhcx1 immunodetection on IL cultures was performed as previously done for LL cultures in
(Fig. S14E in (Giovagnetti et al., 2022)), using a rabbit polyclonal anti-LHCSR antibody from
*Chlamydomonas* (gifted from K. K. Niyogi, University of California, Berkeley, USA, dilution 1: 5 000)
which can detect all of *P. tricornutum* Lhcx isoforms (Bailleul *et al.*, 2010; Taddei *et al.*, 2016, 2018)."

- The reference list needs deep revision; for example, ref nr 3 and nr 34 need the text reviewed.
Some references include the DOI, others do not. Reference nr 48 is a 2022 publication. Line 323:

**“...potential epistatic control of Lhcx1 over xanthophyll synthesis pathway, similar to the control**
**PsbS exerts over zeaxanthin in Arabidopsis,” while ref. nr 56 says other, I believe.**

>> It seems that there was problem with citation generating software compatibility between computers.
We thank **Reviewer 1** for thoroughly revising the reference list and pointing this out. We have corrected
these mistakes and verified all the reference list.

**- Line 362, could the sigmoidal-like relationship between NPQ/qZ and DES, during the rapid**
**relaxation of both NPQ and qZ, be an index of a more complex NPQ model, and suggest the**
**coexistence of other quenching mechanisms, possibly related to other components such as lumen**
**acidification or transthylakoid pmf? Components that, when considering only the kinetics of qZ,**
**might be underestimated? This might affect the main conclusion of the manuscript, i.e. P.**
**tricornutum as a natural reductionist system.**

Before addressing this comment, we would like to apologize for using the inappropriate term “sigmoidal”
here. A sigmoidal curve would imply that the NPQ lags at low DES (which is the case) and reach an
asymptotic value at high DES (which is not observed in our dataset). We have changed “the sigmoidal-like
relationships between qZ and DES for “the deviation from proportionality between qZ and DES”. In the
Results section, we now explain the deviation like this:

L225-227: **“For IL strains, NPQ relaxation in darkness was faster than in LL, and the NPQ/DES relationship**
**deviated from strict proportionality for some strains, showing a trajectory that did not pass through the**
**origin of the axes.”**

That said, the comment of this reviewer is correct: the persistence of DT after complete NPQ relaxation in
some IL-grown strains is consistent with observations in plants, where NPQ relaxation precedes zeaxanthin
epoxidation due to the additional role of PsbS as pH sensor. We understand the reviewer’s consideration of
this hypothesis. We first want to emphasize that this non-proportionality between NPQ and DES is rarely
documented in the literature (limited to reports by Lavaud *et al.*, 2002, FEBS, 2012) and is also rare in our
dataset (appearing significant only in LtpM, LtpY and LtpW strains grown in IL, as shown in Fig S5).

To explain these rare cases of deviation from proportionality, we favor our “multiple xanthophyll pools”
hypothesis over the “multiple quenching mechanisms/pH sensing” hypothesis proposed by this reviewer
for two reasons. First, our hypothesis is more parsimonious: the existence of different xanthophyll pools is
established in *P. tricornutum* (Schumann *et al.*, 2007; Lepetit *et al.*, 2010; Lavaud *et al.*, 2012), whereas
this potential second quenching mechanism/pH sensing remains purely hypothetical, especially since the
observations of Buck *et al.*, 2021 and Giovagnetti *et al.*, 2022 strongly suggested that Lhcx1 does not
function as a pH sensor. Second, only our hypothesis is in line with the comparison between LL and IL
conditions. Indeed, if the deviations observed in IL-grown strains were due to an additional quenching
mechanism or pH sensing, this process would have to be faster than qZ relaxation (DT epoxidation) in IL.
In this case, we would expect an even larger deviation under LL conditions, where qZ relaxation (DT
epoxidation) slows down by about twofold (see Fig S4), although the phenomenological relationship
between NPQ, DES, and Lhcx1 remains stable. Conversely, the “multiple xanthophyll pools” hypothesis
would predict the observed stronger deviation in IL than in LL, because the faster DT epoxidation in IL
makes it more difficult to equilibrate DES between xanthophyll pools, keeping pace with qZ relaxation.

We added, L420-426: **“We propose that this explains the deviation from proportionality between qZ and**
**DES in some of the strains grown in IL (Fig. S5), consistent with results from (Lavaud *et al.*, 2012,**
**Lavaud *et al.*, 2002, FEBS). In some IL-grown strains, some DT remains when NPQ is fully relaxed (Fig**
**S5), likely indicating that the qZ-involved xanthophyll pool undergoes faster epoxidation than the lipid-**
**phase soluble one. This interpretation is also supported by the absence of deviation in LL-grown cells,**

where qZ relaxation (DT epoxidation) is two-fold slower, allowing more time for DES equilibration
between the xanthophyll pools.”

In summary, we believe our SV-lake model offers the simplest and most effective explanation of the data.
Although future research will need to challenge this model under different growth conditions and light
protocols, and it is possible that situations arise where the model does not fully apply, its current scope of
validity is already broad (Fig 3h contains 291 experimental points, each combining measurements of NPQ,
DES and Lhcx1 accumulation. This model encompasses a wide range of Lhcx1 levels, xanthophyll
concentrations, DES values, and easily implementable growth conditions. For a natural reductionist system,
it is essential to have a robust but realistic scope of application, without the need for infinite generalizability.

Minor points:

- I think the color code in the figures could also be accompanied by the name of the line used in that
panel. Having the genotypes immediately evident in the panel, rather than searching through long
captions, simplifies reading and understanding.

We agree with the reviewer that this would help the reader, when a small number of strains is used.
However, we reasoned that it could have the opposite effect of making the panels difficult to read if the
number of strains is too important. Therefore, we added the genotypes on the panel when showing three
strains or less (Fig. 3a-f, Fig. 4a-b and Fig. 6a-b), and listed them in the figure’s legend when more than 3
strains are used.

- **Figure S1: why is there so much diversity in the optical densities of the D2 bands of different**
**replicates?**
- **In the methods, it is not indicated how the total proteins content of the samples used for the**
**western blot were determined.**

The D2 bands showed considerable variation between replicates, especially between replicates 1 and 2.
We observed similar variability for the ATP synthase β subunit as well. This variation could stem from
differences in loading amounts, transfer efficacy or exposure settings on the Bio-Rad system. We loaded
similar protein amount. Total proteins were extracted using SDS lysis buffer as described in Taddei et al,
and quantified using Pierce BCA Protein Assay kit. However, it’s well known that the relationship
between obtained signals (optical density of bands) and protein amount is often not strictly linear. To
reduce the inevitable biases associated with Western blot quantification, we normalized Lhcx1 using the
combined signal of two loading controls: ATP synthase β subunit and D2. This normalization approach
yielded a consistent Lhcx1 range across the fourteen strains in different replicates.

- **Line 170, (Fig. 1d-g)**

This was corrected.

**Reviewer #2 (Remarks to the Author):**

The work from Croteau et al. aims at investigating the NPQ mechanisms in the diatom *Phaeodactylum*
*tricornutum*, using an interesting quantitative approach to distinguish between different hypotheses.
The motivation of the work is also to expand the use of an eukaryotic algae, not belonging to the green

lineage. This is highly relevant for the field, as it is becoming evident that regulation of photosynthesis
mechanisms is different in diverse organisms and that results obtained in models such as Arabidopsis and
Chlamydomonas cannot be generalized, differently from what is often done in the literature.
The work presents a solid and data-based approach to analyse NPQ of these organisms and is a valuable
contribution to the field.

We thank the reviewer for this comment and are pleased to see our shared perspective on the need to
consider variations in photosynthesis regulation across diversity.

**Comments:**

**1. Lines 108 and the following from the Introduction. This part is a bit hard to follow, especially for**
**non-experts. In a journal targeted to a wide audience, an effort to explain better some concepts**
**from the field should be taken.**

**e.g., lines 112 the concept of ‘economic quenching’ in plants should be better described. Since this is**
**also in the title of the present manuscript, it should not be left to a reference to previous literature.**

We are committed to making this manuscript as accessible and engaging as possible for a wide readership.
That’s why we decided to provide a detailed explanation of the SV-lake model’s intricacies in the
appendix (Text S1), while keeping specialized terminology to a minimum in the main text. In Text S1, we
also demonstrate the alignment of our results -interpreted within the SV-lake model framework- with
those from previous studies that gave rise to alternative models (Lavaud et al., 2002, Plant Phy,
Giovagnetti & Ruban 2017, BBA, Buck et al., 2019, Nat Comm). References to Text S1 have been added
in the introduction (L152, 158, 190). Additionally, we included a section to clarify our perspective on
“economic quenching”:

L134-136: “Due to the expected proportionality between σPSII and F_V'/F_M' , the different observations
(both σPSII and F_V'/F_M' higher than predicted) likely indicate a shared phenomenon (see Text S1) that
seems *a priori* inconsistent with the SV-lake model.”

Regarding the “economic quenching”, there must be a misunderstanding here: this concept is neither in
the title of this manuscript nor a central concept in this work. We mention it as one of the models
proposed in the past in response to the apparent disagreement between experimental and theory based on
SV-lake model. From our understanding, the “economic quenching” means that NPQ quenching
efficiency would be more efficient for open than closed reaction centres (see Belgio et al., 2014, Nat
Comm, Giovagnetti & Ruban, 2017, BBA). We added a section to clarify our point:

L121-129: “Although the linear relationship between NPQ and DT (Equation 1) still holds, deviations
from theory were reported regarding the relationship between NPQ and PSII photochemistry (Equation
2). In (Lavaud *et al.*, 2002, Plant Phy), the authors measured a larger than predicted non-photochemical
quenching of minimal fluorescence (F_0' , open PSII reaction centres), relative to the non-photochemical
quenching of maximal fluorescence (F_M' , closed reaction centres) (see Text S1). In (Giovagnetti &
Ruban, 2017), a higher than predicted F_V'/F_M' for a given NPQ, or an apparent “excess of PSII
photochemistry”, was reported. In both scenarios, the efficiency of the non-photochemical quencher
appeared higher when reaction centres were open rather than closed, a pattern previously coined
“economic quenching” in plants (Belgio *et al.*, 2014).”

We have also significantly revised the Text S2 to more clearly demonstrate how our "heterogeneity
hypothesis" satisfactorily explains our experimental results, without the need to invoke the concept of
"economic quenching."

**2. Lines 125 ‘Another argument against the original idea that DT is the SV-quencher is the strict**
**requirement of a second molecular effector for NPQ: LhcX proteins 45,46, which are LHC stress**
**related sub-family ... ‘. I find this part is a bit confusing. Assuming that a binding of DT to LHCX**
**is the more parsimonious hypothesis to explain the data. However, DT bound to LHCX could still**
**be the quencher; only it would require the protein binding.**

We agree with **Reviewer 2** that the most parsimonious hypothesis is that DT binds to, or at least interacts
with, LhcX to form a Stern-Volmer quencher. In this section, we wanted to refer to the original hypothesis
that de-epoxidized xanthophylls *by itself* plays the role of Stern-Volmer quencher (regardless of their
binding partner), tested for DT by Lavaud et al., 2002, Plant Phy, and also for zeaxanthin in plants Bilger
& Bjorkman, 1990, Photosynth Res, (among others). Our formulation was indeed a little bit confusing.
We have modified these two sentences:

L137-138: “Another challenge concerns the nature of the SV-quencher, which cannot be DT *alone* since
the slope of the NPQ vs. DT relationship is highly variable”

L144-145: “Another argument against the original idea that DT *alone* acts as an SV-quencher (Bilger &
Björkman, 1990; Lavaud *et al.*, 2002, Plant Phy) is the strict requirement of a second molecular effector
for NPQ: LhcX proteins”

**3. Line 218. One of the conclusions is to exclude that there is a ‘direct pH regulation of**
**LHCSR3/PsbS as in green organisms 42,43’. I recommend some caution with this conclusion.**
**Although LHCX alone is not inducing quenching and clearly DT is essential for NPQ activation, it**
**should be remembered that activation of the xanthophyll cycle also depends on luminal pH. The**
**possibility that quenching requires low pH activation of LHCX1 cannot be excluded. This is a**
**possibility, assuming that LHCX1 is activated at the same time or after DT synthesis. Since VDE in**
**diatoms is activated at less acid pH than in plants, according to some old literature, this is a realistic**
**hypothesis.**

We thank **Reviewer 2** for pointing out that our reasoning needed clarification. We hope that our answers
to **Reviewer 1**’s first and last comments, together with the additional passages we have included, will
provide a satisfying explanation of our reasoning and conclusion. As mentioned in our response to
**Reviewer 1**, we believe that the works of (Buck et al, 2021) and (Giovagnetti et al, 2021) provide
compelling evidence again the involvement of pH sensing by LhcX1 in NPQ, either under low pH or a
more neutral pH activation.

That said, we would like to address in more detail the alternative proposal of **Reviewer 2**, as we have also
considered this possibility in the past. We agree that an alternative view, where (i) LhcX1 senses pH via a
novel mechanism not involving the lumen-exposed acidic residues, and (ii) LhcX1 pH-activation relaxes
more slowly than the VDE pH-activation during NPQ relaxation, could also be consistent with our data.
However, to remain consistent with the well-documented proportionality between NPQ and DES in this
study and in the broader literature about pennate diatoms -including induction, steady-state and relaxation
phases of NPQ- this alternative would also require that (iii) LhcX1 pH-activation initiates more rapidly
than the VDE pH-activation during NPQ generation in all species, mutants, and growth conditions tested
to date.

In other words, this hypothesis posits that a pH-dependent mechanism at LhcX1 is essential for NPQ but
never affects its magnitude or kinetic changes – it is in practice invisible. This alternative hypothesis is
not only *considerably less parsimonious* (given that there is currently no support for hypothesis (i), (ii)

and (iii)), but it is also *not experimentally testable*. Thus, we believe that the SV-lake model, where the
Stern-Volmer quencher is DT interacting with Lhcx1 in PSII, remains the simplest and most
comprehensive model to explain our results, as well as previous data in the literature.

Minor point

**Line 48 “(PS) have similar absorption spectra and do not form supercomplexes (see forthcoming**
**review (Croteau et al.,’, the citation is missing**

This was corrected.

**Reviewer #3 (Remarks to the Author):**

The authors have provided a very thorough and useful analysis of the LHCX/DT role in NPQ in diatoms,
from which the field will surely benefit.

We would like to thank the reviewer for the very positive assessment of our work.

I have a single major point that required discussion/revisions from the authors prior resubmission,
however.

**Both in the abstract, text and discussion, the authors mention that there is likely binding of DT to**
**LHCX (as for example stated in the abstract). This is not however based on any structural evidence,**
**isolation of LHCSX or even reconstitution with pigments, neither in this work nor in the past, to my**
**understanding. For example, in plants it was shown that zeaxanthin does not bind to any LHC,**
**even if pigment binding sites might be available from a structural point of view (Xu et al Scientific**
**Reports 2015).**

**All these points should be rediscussed, at the moment I had the impression that it seems that from**
**this work we could conclude that there is binding which instead we cannot conclude based on the**
**presented data. If this point was crucial for some of their analysis, the authors should explain**
**whether their results/conclusions will hold also in case DT does not bind to LHCX. If instead this is**
**only a speculation, it should be more carefully rephrased in all the points given the lack of evidence.**

We agree that we overreached with this hypothesis in some instances and thank **Reviewer 3** for pointing
this out and for mentioning the work about zeaxanthin (Xu et al., 2015). **Reviewer 3** is right that Buck et
al., Plant J 2021 suggest potential pigment binding sites from a structural point of view, but did not prove
binding of DT to any Lhex isoform.

However, the main conclusions of our work, i.e., 1) the fast-relaxing NPQ behaves as a quenching by a
Stern-Volmer homogeneous quencher whose amount is proportional to Lhcx1 * DES, 2) that the
photosynthetic unit is efficiently described by the lake model in pennate diatoms, provided that
heterogeneity is taken into account, and 3) that those properties make diatoms a convenient natural
reductionist system, do not depend on our interpretation of structural insights from (Buck et al, 2021). It
influences only the nature of the SV quencher, which we can only speculate based on the existing
literature and the robust relationship between NPQ DES and Lhex1.

The observations that NPQ is proportional to DES in *P. tricornutum*, in the presence of the motif
identified in (Buck et al., 2021), and that regardless of DES, NPQ is suppressed in the absence of this
motif, clearly indicates that the potential binding site modulates the quenching efficiency of DT. In line
with the suggestion of the reviewer, we modified the text to clarify the fact that the findings of (Buck et

al, 2021) support that NPQ requires an interaction between Lhcx1 and DT of yet undetermined nature,
and to clarify the fact that DT binding to Lhcx1 is only a hypothesis.

In the Abstract, we have removed “likely through the binding of diatoxanthin on Lhcx1”.

In the results section, L247-250, we changed “Assuming each Lhcx1 protein binds DD/DT (see
(Huysman *et al.*, 2010) and Discussion) and that only the Lhcx1-DT complex generates Q and induces
NPQ/qZ, then $[Q]$ would be the product of $[Lhcx1]$ (the concentration of xanthophyll binding sites) and
DES (the proportion of xanthophylls in the DT form)” for “Assuming each Lhcx1 protein interacts with
DD/DT (possibly via binding see (Buck *et al.*, 2021) and Discussion) and that only the Lhcx1-DT
complex generates Q and induces NPQ/qZ, then $[Q]$ would be the product of $[Lhcx1]$ (the concentration
of xanthophyll interaction sites) and DES (the proportion of xanthophylls in the DT form) (Fig. 3i).”.

The Discussion about the nature of the quencher was also revised notably the following passage:

L400-409: “Nevertheless, Lhcx1-DT binding is not the only possible mechanistic model for which DT
interacting with Lhcx1 could form an SV-quencher. Like suggested for lutein with CP29 in plants
(Accomasso *et al.*, 2024), an Lhcx1 binding pocket could allow for more planar s-trans conformer of the
xanthophyll, of which the S_1 state could form the quencher. Additionally, in *Arabidopsis*, zeaxanthin
accumulates at the antenna periphery to induce quenching, rather than being converted from violaxanthin
strongly protein-bound within the antenna (Xu *et al.*, 2015). The further observation that zeaxanthin-
related quenching occurs in isolated membranes (Gilmore *et al.*, 1998), but not in isolated complex,
suggests that the peripheral zeaxanthin is weakly linked and lost during the purification process (Xu *et al.*,
2015). In contrast, in *P. tricornutum*, isolated FCPs contain both DD and DT (Lepetit *et al.*, 2010) and are
capable of quenching (Nagao *et al.*, 2021).”

**REVIEWERS' COMMENTS**

Reviewer #1 (Remarks to the Author):

The authors have satisfactorily addressed my previous questions and concerns. The manuscript
has been significantly improved during the revision process. I have no further questions and
recommend the publication of the work in Nature Communications.
Luca Dall'Osto

Reviewer #3 (Remarks to the Author):

The authors addressed my comments and I also appreciate their thorough discussion of all
other reviewers' comments.

Thanks to reviewers 1 and 3 for their positive feedbacks.

Reviewer #2 (Remarks to the Author):

In my opinion, authors answered satisfactorily to all reviewers' comments.

I'd only add a minor point on the response to reviewer 1 on NPQ components nomenclature,
since as stated by authors, can be confusing.

In plants the distinction between qE and qZ is made based on the kinetics, with qE being the
fastest component. In motivating their choice to call the NPQ component qZ, I would
recommend making a direct comparison of the diatoms and plants NPQ relaxation kinetics. I
tried to reconstruct this comparison looking into the supplementary material and it was not
straightforward. In this discussion it should also be considered that in plants qE is also
dependent on zeaxanthin. It is not necessary for qE but its presence enhance it.

I also realized that in the whole supplementary material there is not an example of a NPQ
measurement. I understand this is redundant and published many times, but since the
manuscript focuses on these measurements, a few examples in supplementary material could
still be useful for some readers.

Thanks to Reviewer 2 for the positive comment.

We agree that there was no clear example of NPQ measurements and added a supplementary figure
displaying fluorescence kinetics and NPQ/qZ kinetics calculated from the fluorescence traces, in three
strains: the Pt2 wildtype, the Lhcx1 KO and the overexpressor LtpM. This supplementary Figure is now
Fig S3.

We have also modified a section of the discussion about the distinction between qE and qZ to account for
Reviewer 2's comments (note that this section already mentioned the distinction between qE and qZ
based on relaxation kinetics, in plants):

L383-390: "Given that qE traditionally refers to the fast-relaxing NPQ *directly* triggered by luminal pH-
sensing antenna proteins, we conclude that qE does not accurately represent the fast-relaxing NPQ
component in pennate diatoms. Instead, this component aligns more closely with zeaxanthin-dependent

quenching (qZ), first described in *Arabidopsis* PsbS mutants^{21,49}. Indeed, qZ is an NPQ component
displaying slower relaxation than qE, as it strictly follows the conversion of the de-epoxidized pigment
(zeaxanthin) back to the epoxidized form (violaxanthin). In plants, qE is also enhanced by zeaxanthin but
the kinetics of its induction and relaxation differ from those of the xanthophyll cycle.”